# PolicyFlow: Policy Optimization with Continuous Normalizing Flow in Reinforcement Learning

**Shunpeng Yang**[1,4]**, Ben Liu**[2] **& Hua Chen**[3,4†]
[1]Hong Kong University of Science and Technology, [2]Southern University of Science and Technology
[3]Zhejiang University-University of Illinois Urbana-Champaign Institute, [4]LimX Dynamics
[†] Corresponding author, `huachen@intl.zju.edu.cn`

## ABSTRACT

Among on-policy reinforcement learning algorithms, Proximal Policy Optimization (PPO) demonstrates is widely favored for its simplicity, numerical stability, and strong empirical performance. Standard PPO relies on surrogate objectives defined via importance ratios, which require evaluating policy likelihood that is typically straightforward when the policy is modeled as a Gaussian distribution. However, extending PPO to more expressive, high-capacity policy models such as continuous normalizing flows (CNFs), also known as flow-matching models, is challenging because likelihood evaluation along the full flow trajectory is computationally expensive and often numerically unstable. To resolve this issue, we propose PolicyFlow, a novel on-policy CNF-based reinforcement learning algorithm that integrates expressive CNF policies with PPO-style objectives without requiring likelihood evaluation along the full flow path. PolicyFlow approximates importance ratios using velocity field variations along a simple interpolation path, reducing computational overhead without compromising training stability. To further prevent mode collapse and further encourage diverse behaviors, we propose the Brownian Regularizer, an implicit policy entropy regularizer inspired by Brownian motion, which is conceptually elegant and computationally lightweight. Experiments on diverse tasks across various environments including MultiGoal, PointMaze, IsaacLab and MuJoCo Playground show that PolicyFlow achieves competitive or superior performance compared to PPO using Gaussian policies and flow-based baselines including FPO and DPPO. Notably, results on MultiGoal highlight PolicyFlow's ability to capture richer multimodal action distributions.

## 1 INTRODUCTION

Reinforcement learning (RL), particularly policy gradient (PG) methods, has achieved remarkable success in complex sequential decision-making tasks, ranging from robotic control (Andrychowicz et al., 2020; Rudin et al., 2022; Cheng et al., 2024; He et al., 2025) to aligning large language models with human preferences (Ouyang et al., 2022; Zhai et al., 2024; Guo et al., 2025). Among PG methods, Proximal Policy Optimization (PPO) (Schulman et al., 2017) remains a standard due to its simplicity and generally reliable performance, widely used in complex robotic control tasks (Lee et al., 2020; Chen et al., 2025), and recently for fine-tuning generative policies (Black et al., 2023; Ren et al., 2024). PPO optimizes policies via surrogate objectives based on importance ratios, which require nontrivial likelihood evaluation. For tractable computation, policies are typically modeled by Gaussian distribution. While convenient, Gaussian policies are limited in representing complex, multimodal, or highly skewed action distributions, motivating the use of more expressive generative models.

In recent years, generative models, such as diffusion models (Ho et al., 2020; Song et al., 2020) and continuous normalizing flows (Lipman et al., 2022; Tong et al., 2023), have emerged as a powerful class of models capable of capturing complex, multimodal distributions. These models have been successfully applied to imitation learning, where they model policy distributions directly from demonstration data (Chi et al., 2023; Ze et al., 2024), effectively capturing trajectory diversity and

complex behaviors. However, computing importance ratios or likelihoods for these models typically requires iterative ODE/SDE simulations and path-wise backpropagation (Chen et al., 2018), which is computationally expensive and prone to exploding or vanishing gradients. This makes direct application of such models in PPO-style updates slow, memory-intensive, and potentially unstable, limiting their practicality for efficient on-policy reinforcement learning.

Motivated by these challenges, we propose PolicyFlow, a novel on-policy RL algorithm that combines the expressiveness of continuous normalizing flows with a PPO-style clipped objective, enabling efficient and stable policy optimization. Our contributions are as follows:

- *Importance ratio approximation for CNF policies.* PolicyFlow approximates importance ratios by evaluating the variations of CNF's velocity field along interpolation paths, avoiding the costly path-wise backpropagation.

- *Brownian entropy regularization.* We propose a lightweight entropy regularizer inspired by Brownian motion (Einstein, 1905), which promotes monotonic entropy growth, mitigates mode collapse, and encourages diverse actions without explicitly computing the CNF policy's entropy.

These results demonstrate PolicyFlow's potential as a practical and expressive framework for on-policy RL. Code and project page are available at `https://policyflow2026.github.io/`.

## 2 RELATED WORK

### 2.1 FLOW/DIFFUSION-BASED REPRESENTATIONS OF RL POLICIES

Flow and diffusion models provide highly expressive, multi-modal distributions, making them attractive as policy parameterizations in reinforcement learning. Compared to conventional categorical or Gaussian policies, these generative models allow richer action distributions and can potentially capture a broader set of behaviors. In robotics, flow-based and diffusion-based models have been widely adopted as policy representations (Chi et al., 2023; Lei et al., 2025; Intelligence et al., 2025; Gao et al., 2025). However, progress has approached a bottleneck, as these models are typically trained solely through denoising score matching or flow matching on offline datasets, without incorporating reinforcement learning. This limitation has motivated researchers to explore using RL to directly train generative policies.

In **offline RL**, diffusion-based policies have been widely adopted to model complex action patterns from static datasets, often guided by value functions or energy-based objectives (Wang et al., 2022; Lu et al., 2023; Psenka et al., 2023; Zhang et al., 2025). These approaches have achieved strong results on D4RL benchmarks and inspired actor–critic variants that couple generative policies with value estimation (Wang et al., 2024; Fang et al., 2024). To address the heavy sampling and computational demands of diffusion models, recent works have also explored more efficient formulations (Kang et al., 2023).

In **online RL**, the setting is more demanding because it requires efficient sampling, stable importance-ratio estimation, and tractable (or well-approximated) likelihoods. Several works (Wang et al., 2024; Chao et al., 2024; Ding et al., 2025) directly backpropagate policy gradients through the full diffusion/flow chain, which enables end-to-end optimization but risks exploding or vanishing gradients. Practical recipes for fine-tuning expressive diffusion policies with policy-gradient-style updates have also been proposed in DPPO (Ren et al., 2024), which enables structured on-manifold exploration and stable long-horizon training. While effective in fine-tuning scenarios, its performance tends to degrade when training from scratch, as off-manifold exploration becomes necessary. FPO (McAllister et al., 2025) instead estimates policy importance ratios through an ELBO objective, which offers a scalable approximation but introduces asymmetric estimation bias—more reliable when the importance ratio increases than when it decreases—potentially amplifying variance and affecting stability. To mitigate this issue, FPO typically requires larger batch sizes during updates.

Overall, prior work illustrates both the promise and the limitations of expressive generative policies in online RL: while they expand the representable policy class, existing approaches may suffer from unstable optimization, high computational cost, or biased approximations. PolicyFlow is an on-policy algorithm that seeks to address these challenges without backpropagating through full

generative chains, without treating diffusion as an internal Markov Decision Process as in DPPO, and while avoiding the asymmetric bias in FPO.

## 2.2 POLICY ENTROPY REGULARIZATION

Entropy regularization has long been used to encourage exploration and prevent mode collapse in reinforcement learning. Classical approaches show its effectiveness for categorical policies in discrete action spaces (Mnih et al., 2016) and Gaussian policies in continuous control (Haarnoja et al., 2018). Extending this principle to flow-based policies, however, is challenging: action log-likelihoods are generally intractable, making entropy estimation expensive.

In principle, closed-form dynamics of entropy under continuous normalizing flows can be derived via the divergence of the velocity field integrated along the flow path (Chen et al., 2018; Tian et al., 2024). While theoretically sound, this requires costly divergence evaluation and path integration, which limits scalability. More heuristic solutions have also been explored: Wang et al. (2024) approximate entropy using Gaussian mixture models, adjusting the injected noise to diffusion output accordingly, while Ding et al. (2024) inject uniform noise into training samples to artificially inflate entropy. These strategies are effective in specific cases, and can be good choices when additional computational cost is not a concern or when the range of action samples is known in advance.

Our approach introduces an implicit entropy regularizer, inspired by Brownian motion, that directly shapes the velocity field toward entropy-increasing dynamics. This design avoids expensive log-likelihood computation and bypasses the need for ad hoc noise heuristics. Since entropy regularization was not explicitly addressed in methods such as FPO, our regularizer provides a principled and lightweight alternative in flow-based policy optimization.

## 3 BACKGROUND

We consider a Markov Decision Process (MDP) defined by the tuple $(\mathcal{S}, \mathcal{A}, p, r, \gamma)$, where $\mathcal{S}$ is the state space, $\mathcal{A}$ is the action space, $p(\mathbf{s}' \mid \mathbf{s}, \mathbf{a})$ denotes the transition dynamics with state $\mathbf{s} \in \mathcal{S}$ and action $\mathbf{a} \in \mathcal{A}$, $r(\mathbf{s}, \mathbf{a})$ is the reward function, and $\gamma \in [0, 1)$ is the discount factor. The agent's objective is to learn a policy $\pi(\mathbf{a}|\mathbf{s})$ that maximizes the expected cumulative discounted return:

$$J(\pi) = \mathbb{E}_{p(\tau|\pi)} \left[ \sum_{k=0}^{\infty} \gamma^k r(\mathbf{s}_k, \mathbf{a}_k) \right] \tag{1}$$

where $\tau = \{\mathbf{s}_0, \mathbf{a}_0, \mathbf{s}_1, \mathbf{a}_1, ...\}$ denotes a trajectory sampled from the environment under policy $\pi$.

Among policy gradient algorithms, PPO has become one of the most widely adopted due to its simplicity and empirical stability. PPO optimizes the policy by maximizing a clipped surrogate objective (Schulman et al., 2017):

$$J^{\text{PPO}}(\pi) = \mathbb{E}_{p_{\hat{\pi}}(\mathbf{s})} \mathbb{E}_{\hat{\pi}(\mathbf{a}|\mathbf{s})} \left[ \min \left( \frac{\pi(\mathbf{a}|\mathbf{s})}{\hat{\pi}(\mathbf{a}|\mathbf{s})} A_{\hat{\pi}}(\mathbf{s}, \mathbf{a}), \text{clip} \left( \frac{\pi(\mathbf{a}|\mathbf{s})}{\hat{\pi}(\mathbf{a}|\mathbf{s})}, 1 - \epsilon, 1 + \epsilon \right) A_{\hat{\pi}}(\mathbf{s}, \mathbf{a}) \right) \right] \tag{2}$$

where $\hat{\pi}(\mathbf{a}|\mathbf{s})$ is a reference policy, $p_{\hat{\pi}}(\mathbf{s})$ is policy's state distribution (Schulman et al., 2015), $A_{\hat{\pi}}(\mathbf{s}, \mathbf{a})$ is the corresponding advantage function, and the clipping range $\epsilon$ is a small positive hyperparameter (typically in the range $[0.1, 0.3]$) that controls the maximum allowable deviation of the likelihood ratio from one. This formulation prevents the updated policy from deviating too far from the reference policy, thereby ensuring more stable learning in practice.

More recently, Frans et al. (2025) showed that PPO and related algorithms can also be interpreted under a proxy objective of the form:

$$\hat{J}(\pi) = \mathbb{E}_{p_{\hat{\pi}}(\mathbf{s})} \mathbb{E}_{\pi(\mathbf{a}|\mathbf{s})} A_{\hat{\pi}}(\mathbf{s}, \mathbf{a}). \tag{3}$$

As long as the divergence between the resulting policy $\pi^* = \arg\max_\pi \hat{J}(\pi)$ and the reference policy $\hat{\pi}$ remains bounded optimizing this proxy objective guarantees monotonic improvement of the true objective, i.e., $J(\pi^*) > J(\hat{\pi})$.

# 4 POLICY OPTIMIZATION WITH CONTINUOUS NORMALIZING FLOW

To overcome the limitations of Gaussian parameterizations, we propose to represent policies using continuous normalizing flows. Specifically, we define a conditional flow $\varphi : [0,1] \times \mathbb{R}^d \times \mathbb{R}^n \to \mathbb{R}^d$ governed by the ordinary differential equation (ODE):

$$\frac{d}{dt}\varphi_t(\mathbf{z};\mathbf{s}) = v_t(\varphi_t(\mathbf{z};\mathbf{s});\mathbf{s}), \quad \varphi_0(\mathbf{z};\mathbf{s}) = \mathbf{z} \tag{4}$$

where $\mathbf{z} \in \mathbb{R}^d$ is a latent variable, $\mathbf{s} \in \mathbb{R}^n$ is the state and $v : [0,1] \times \mathbb{R}^d \times \mathbb{R}^n \to \mathbb{R}^d$ is a time-dependent velocity field which can be parameterized by a neural network.

Similar to Wang et al. (2024), the policy generates actions by integrating the flow to its terminal time and adding Gaussian noise:

$$\mathbf{a} = \varphi_1(\mathbf{z};\mathbf{s}) + \mathbf{n}, \quad \mathbf{z} \sim p_z(\mathbf{z}), \quad \mathbf{n} \sim \mathcal{N}(\mathbf{n};\mathbf{0},\boldsymbol{\sigma}^2). \tag{5}$$

Here, the injected noise $\mathbf{n}$ not only facilitates exploration but also ensures compatibility with the PPO-style surrogate objective, allowing us to naturally extend PPO's original formulation to continuous normalizing flows. While various choices are possible for $p_z(\mathbf{z})$ in principle, we follow common practice in generative model and choose a standard Gaussian distribution, namely $p_z(\mathbf{z}) = \mathcal{N}(\mathbf{z};\mathbf{0},\mathbf{1})$.

This construction induces a policy distribution

$$\pi(\mathbf{a}|\mathbf{s}) = \int \pi(\mathbf{a}|\mathbf{z},\mathbf{s})p_z(\mathbf{z})\,\mathrm{d}\mathbf{z}, \quad \pi(\mathbf{a}|\mathbf{z},\mathbf{s}) = \mathcal{N}(\mathbf{a};\varphi_1(\mathbf{z};\mathbf{s}),\boldsymbol{\sigma}^2). \tag{6}$$

This representation is strictly more expressive than a conventional Gaussian policy, as it can recover simple unimodal Gaussian distributions while also modeling arbitrarily complex, multimodal, or highly non-Gaussian action distributions.

Under this parameterization, the policy proxy objective in Eq. (3) can be rewritten as

$$\hat{J}(\pi) = \mathbb{E}_{p_{\hat{\pi}}(\mathbf{s}),p_z(\mathbf{z})}\mathbb{E}_{\pi(\mathbf{a}|\mathbf{z},\mathbf{s})}A_{\hat{\pi}}(\mathbf{s},\mathbf{a}) = \mathbb{E}_{p_{\hat{\pi}}(\mathbf{s}),p_z(\mathbf{z})}\mathbb{E}_{\hat{\pi}(\mathbf{a}|\mathbf{z},\mathbf{s})}\left[\frac{\pi(\mathbf{a}|\mathbf{z},\mathbf{s})}{\hat{\pi}(\mathbf{a}|\mathbf{z},\mathbf{s})}A_{\hat{\pi}}(\mathbf{s},\mathbf{a})\right], \tag{7}$$

which explicitly connects the flow-based policy representation with the standard objective used in policy optimization.

In principle, the importance ratio can be computed by simulating both flows $\varphi$ and $\hat{\varphi}$ through their ODEs during training. However, this is often computationally expensive and numerically unstable, as neural ODEs may suffer from exploding/vanishing gradients or high memory usage during training. Next, we describe an alternative objective that avoids directly simulating the ODEs to compute this importance ratio during training.

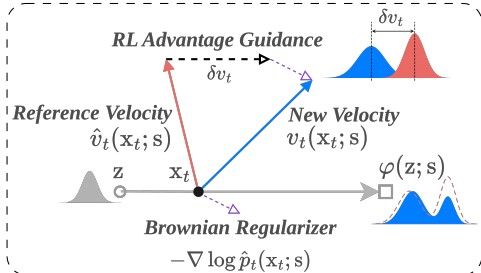

Now let $p_n(\cdot\,;\boldsymbol{\mu},\boldsymbol{\sigma}^2)$ denote the Gaussian density function with mean $\boldsymbol{\mu}$ and variance $\boldsymbol{\sigma}^2$. A key observation is that the likelihood ratio between Gaussian distributions is shift-invariant, which means

$$\frac{\pi(\mathbf{a}|\mathbf{z},\mathbf{s})}{\hat{\pi}(\mathbf{a}|\mathbf{z},\mathbf{s})} = \frac{p_n\left(\mathbf{a};\varphi_1(\mathbf{z};\mathbf{s}),\boldsymbol{\sigma}^2\right)}{p_n\left(\mathbf{a};\hat{\varphi}_1(\mathbf{z};\mathbf{s}),\hat{\boldsymbol{\sigma}}^2\right)} = \frac{p_n\left(\mathbf{a}-\hat{\varphi}_1(\mathbf{z};\mathbf{s});\delta_{\varphi_1}(\mathbf{z};\mathbf{s}),\boldsymbol{\sigma}^2\right)}{p_n\left(\mathbf{a}-\hat{\varphi}_1(\mathbf{z};\mathbf{s});\mathbf{0},\hat{\boldsymbol{\sigma}}^2\right)} \tag{8}$$

where $\delta_{\varphi_1}(\mathbf{z};\mathbf{s}) = \varphi_1(\mathbf{z};\mathbf{s}) - \hat{\varphi}_1(\mathbf{z};\mathbf{s})$. Directly computing $\delta_{\varphi_1}(\mathbf{z};\mathbf{s})$ requires simulating the ODEs, which is computationally costly. To alleviate this, we approximate the terminal shift using the velocity variation along the following linear interpolation path:

$$\mathbf{x}_t = (1-t)\mathbf{z} + t\hat{\varphi}_1(\mathbf{z};\mathbf{s}), \quad t \in [0,1]. \tag{9}$$

This approximation replaces the integral over the reference trajectory with an expectation over $t$ along the interpolation path, yielding:

$$\frac{\pi(\mathbf{a}|\mathbf{z},\mathbf{s})}{\hat{\pi}(\mathbf{a}|\mathbf{z},\mathbf{s})} \approx \mathbb{E}_{p(t)}\left[\frac{p_n\left(\mathbf{a}-\hat{\varphi}_1(\mathbf{z};\mathbf{s});\delta_{v_t}(\mathbf{x}_t;\mathbf{s}),\boldsymbol{\sigma}^2\right)}{p_n\left(\mathbf{a}-\hat{\varphi}_1(\mathbf{z};\mathbf{s});\mathbf{0},\hat{\boldsymbol{\sigma}}^2\right)}\right] \tag{10}$$

where $p(t) = U[0,1]$ and the velocity field variation $\delta_{v_t}(\mathbf{x}_t;\mathbf{s}) = v_t(\mathbf{x}_t;\mathbf{s}) - \hat{v}_t(\mathbf{x}_t;\mathbf{s})$.

**Remark (Approximation Error Bound)** *Theoretical analysis (see Appendix A for details) shows that this interpolation-based approximation introduces only a first-order error in the log under small update regimes, which can be naturally enforced by the clipping range $\epsilon$ in PPO.*

$$\left| \mathbb{E}_{p(t)} \left[ \frac{p_n \left( \mathbf{a} - \hat{\varphi}_1(\mathbf{z}; \mathbf{s}); \delta_{v_t}(\mathbf{x}_t; \mathbf{s}), \boldsymbol{\sigma}^2 \right)}{p_n \left( \mathbf{a} - \hat{\varphi}_1(\mathbf{z}; \mathbf{s}); \mathbf{0}, \hat{\boldsymbol{\sigma}}^2 \right)} \right] - \frac{p_n \left( \mathbf{a} - \hat{\varphi}_1(\mathbf{z}; \mathbf{s}); \delta_{\varphi_1}(\mathbf{z}; \mathbf{s}), \boldsymbol{\sigma}^2 \right)}{p_n \left( \mathbf{a} - \hat{\varphi}_1(\mathbf{z}; \mathbf{s}); \mathbf{0}, \hat{\boldsymbol{\sigma}}^2 \right)} \right| = \mathcal{O}(\epsilon). \quad (11)$$

*Importantly, this approach allows us to avoid simulating the full flow trajectory and propagating gradients along it during training, thereby maintaining computational efficiency comparable to PPO with Gaussian policy.*

Now, the velocity field $v$ is parameterized by a neural network with parameters $\theta$. Finally, similar to PPO, we adopt a clipped surrogate-style objective to stabilize training:

$$J^{\text{Flow}}(\theta, \boldsymbol{\sigma}) = \mathbb{E}_{p_{\hat{\pi}}(\mathbf{s}), p_z(\mathbf{z})} \mathbb{E}_{\hat{\pi}(\mathbf{a}|\mathbf{z}, \mathbf{s}), p(t)} \left[ \min \left( \rho A_{\hat{\pi}}(\mathbf{s}, \mathbf{a}), \text{clip} \left( \rho, 1 - \epsilon, 1 + \epsilon \right) A_{\hat{\pi}}(\mathbf{s}, \mathbf{a}) \right) \right] \quad (12)$$

with approximate importance ratio

$$\rho = \frac{p_n \left( \mathbf{a} - \hat{\varphi}_1(\mathbf{z}; \mathbf{s}); \ v_t(\mathbf{x}_t; \mathbf{s}, \theta) - \hat{v}_t(\mathbf{x}_t; \mathbf{s}), \boldsymbol{\sigma}^2 \right)}{p_n \left( \mathbf{a} - \hat{\varphi}_1(\mathbf{z}; \mathbf{s}); \mathbf{0}, \hat{\boldsymbol{\sigma}}^2 \right)}. \quad (13)$$

Thus, simulation of the ODE is only required during sampling (to compute $\hat{\varphi}_1(\mathbf{z}; \mathbf{s})$), while the training objective can be efficiently estimated along the interpolation path with velocity filed variations, without simulating the ODE or backpropagating through the simulated flow trajectories in training.

## 4.1 POLICY ENTROPY REGULARIZATION

Policy entropy maximization is a long-standing technique to encourage exploration and mitigate mode collapse in reinforcement learning. To tackle the difficulties of entropy regularization for flow-based policies, as discussed in Sec. 2.2, we propose a novel entropy regularizer inspired by Brownian motion. Our method differs from prior entropy regularization: instead of explicitly computing policy entropy or heuristically injecting noise, we directly regulate the velocity field to follow an entropy-increasing process. This perspective allows us to avoid costly log-likelihood evaluation while still encouraging diverse exploration, shown as Fig. 1.

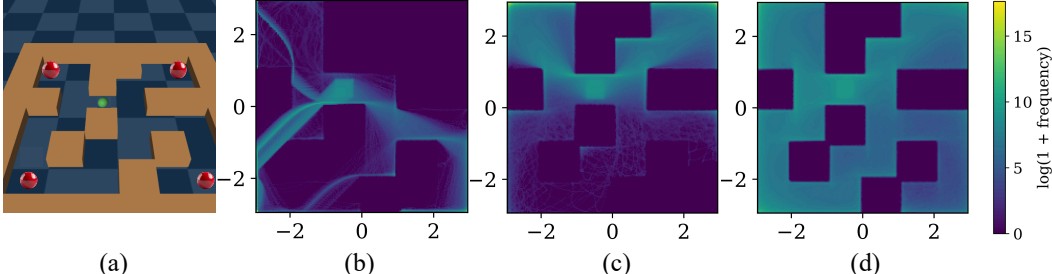

Figure 1: (`PointMaze-Medium-Diverse-GDense-v3`) Exploration Density Maps. (a) Environment overview: the agent is initialized at the green point for each episode, and the four red points indicate goal locations with equal rewards. (b) Exploration heatmap of PPO, showing limited coverage due to the simple Gaussian policy. (c) Exploration heatmap of PolicyFlow without the Brownian regularizer, which improves coverage but still leaves some regions under-explored. (d) Exploration heatmap of PolicyFlow with the Brownian regularizer, achieving near-complete coverage of all feasible locations.

In Brownian dynamics, particles naturally spread toward a uniform distribution, and entropy monotonically increases during the process. Although Brownian motion is defined by a stochastic differential equation, its probability path follows the classic heat equation $\partial p_t(\mathbf{x})/\partial t = \nabla_{\mathbf{x}}^2 p_t(\mathbf{x})$ (Jordan et al., 1998). This equation connects directly to the continuity equation $\partial p_t(\mathbf{x})/\partial t = -\nabla_{\mathbf{x}} \cdot (p_t(\mathbf{x}) v_t(\mathbf{x}))$. By choosing $v_t(\mathbf{x}) = -\nabla_{\mathbf{x}} \log p_t(\mathbf{x})$, the continuity equation recovers the heat equation, showing that entropy growth can be enforced via a carefully shaped velocity field aligned with the negative score.

In practice, to promote flow trajectories that expand like Brownian motion rather than collapse, we want our learned velocity field to follow the negative score of a reference flow. To obtain this score

---

**Algorithm 1** PolicyFlow

---

1: **Input:** initial velocity field parameters $\theta_0$, initial noise variance $\boldsymbol{\sigma}_0^2$, initial value function parameters $\phi_0$

2: **for** iteration $i = 0, 1, 2, \ldots$ **do**

3:     Set reference parameters $\hat{\theta} \leftarrow \theta_i$, $\hat{\boldsymbol{\sigma}}^2 \leftarrow \boldsymbol{\sigma}_i^2$. The reference velocity field is $\hat{v} = v_{\hat{\theta}}$

4:     Collect a set of trajectories $\mathcal{D}_i$ using the reference policy $\pi_{\hat{\theta}, \hat{\boldsymbol{\sigma}}}$

5:     **for** each MDP step $k$ with state $\mathbf{s}_k$ **do**

6:         Sample latent variable $\mathbf{z}_k \sim p_z(\mathbf{z})$ then $\hat{\varphi}_0 = \mathbf{z}_k$

7:         Compute $\boldsymbol{\varphi}_k = \hat{\varphi}_1(\mathbf{z}_k; \mathbf{s}_k)$ by simulating the ODE $\frac{d}{dt}\hat{\varphi}_t = \hat{v}_t(\hat{\varphi}_t; \mathbf{s}_k)$ from $t = 0$ to 1

8:         Sample noise $\mathbf{n}_k \sim \mathcal{N}(\mathbf{0}, \hat{\boldsymbol{\sigma}}^2)$ and form action $\mathbf{a}_k = \boldsymbol{\varphi}_k + \mathbf{n}_k$

9:         Execute $\mathbf{a}_k$, observe next state $\mathbf{s}_{k+1}$ and reward $r_k$

10:        Store transition $(\mathbf{s}_k, \mathbf{a}_k, r_k, \mathbf{s}_{k+1}, \mathbf{z}_k, \boldsymbol{\varphi}_k)$ in $\mathcal{D}_i$

11:     **end for**

12:     For each step $k$, compute rewards-to-go $\hat{R}_k$ and advantage estimates $\hat{A}_k$ using GAE

13:     **for** epoch $= 1, \ldots, E$ **do**

14:         **for** each mini-batch of transitions $(\mathbf{s}_k, \mathbf{a}_k, \hat{A}_k, \mathbf{z}_k, \boldsymbol{\varphi}_k)$ from $\mathcal{D}_i$ **do**

15:             Sample $t_k \sim U[0, 1]$

16:             Or sample $t_k$ from the discrete time points used for numerical simulation of flow ODE

17:             Compute interpolation point $\mathbf{x}_{t_k} = (1 - t_k)\mathbf{z}_k + t_k\boldsymbol{\varphi}_k$

18:             Compute approximate importance ratio

$$\rho_k = p_n\left(\mathbf{a}_k - \boldsymbol{\varphi}_k; v_{t_k}(\mathbf{x}_{t_k}; \mathbf{s}_k, \theta) - \hat{v}_{t_k}(\mathbf{x}_{t_k}; \mathbf{s}_k), \boldsymbol{\sigma}^2\right) / p_n\left(\mathbf{a}_k - \boldsymbol{\varphi}_k; \mathbf{0}, \hat{\boldsymbol{\sigma}}^2\right)$$

19:             Compute the clipped surrogate objective for the mini-batch by

$$J^{\text{Flow}} = \mathbb{E}_k\left[\min(\rho_k\hat{A}_k, \text{clip}(\rho_k, 1 - \epsilon, 1 + \epsilon)\hat{A}_k)\right]$$

20:             Compute Brownian regularizer vector

$$\eta_{t_k}(\mathbf{x}_{t_k}; \mathbf{s}_k, \theta) = (1 - t_k)v_{t_k}(\mathbf{x}_{t_k}; \mathbf{s}_k, \theta) - (\mathbf{x}_{t_k} - t_k\,\hat{v}_{t_k}(\mathbf{x}_{t_k}; \mathbf{s}))$$

21:             Compute the regularization term for the mini-batch by

$$J^{\text{Reg}} = \mathbb{E}_k\left[-w_b\|\eta_{t_k}(\mathbf{x}_{t_k}; \mathbf{s}, \theta)\|_2^2 + \frac{w_g}{2}\sum_{i=1}^d \log(2\pi e \sigma_i^2)\right]$$

22:             Update policy parameters $(\theta_{i+1}, \boldsymbol{\sigma}_{i+1}) = \arg\max_{\theta, \boldsymbol{\sigma}}(J^{\text{Flow}} + J^{\text{Reg}})$.

23:             Update value function parameters by minimizing the mean-squared error:

$$\phi_{i+1} = \arg\min_{\phi} \mathbb{E}_k\left(\mathcal{V}_\phi(\mathbf{s}_k) - \hat{R}_k\right)^2$$

24:         **end for**

25:     **end for**

26: **end for**

---

function from the reference velocity field, we can leverage the result of Liu et al. (2025), the velocity field and score function are explicitly related:

$$\nabla_\mathbf{x} \log \hat{p}_t(\mathbf{x}_t; \mathbf{s}) = \frac{1}{1 - t}(t\,\hat{v}_t(\mathbf{x}_t; \mathbf{s}) - \mathbf{x}_t). \tag{14}$$

Building on this connection, we purpose a practical entropy regularizer:

$$J^{\text{Reg}}(\theta, \boldsymbol{\sigma}) = \mathbb{E}_{p_{\hat{\pi}}(\mathbf{s}), \, p_z(\mathbf{z}), \, p(t)}\left[-w_b\|\eta_t(\mathbf{x}_t; \mathbf{s}, \theta)\|_2^2 + \frac{w_g}{2}\sum_{i=1}^d \log(2\pi e \sigma_i^2)\right], \tag{15}$$

where $w_b, w_g \geq 0$ are tunable coefficients and

$$\eta_t(\mathbf{x}_t; \mathbf{s}, \theta) = (1 - t)v_t(\mathbf{x}_t; \mathbf{s}, \theta) - (\mathbf{x}_t - t\,\hat{v}_t(\mathbf{x}_t; \mathbf{s})). \tag{16}$$

The first term (termed Brownian regularizer) in Eq. (15) encourages the learned velocity field to align with the negative score of the reference flow, promoting expansion of trajectories and preventing collapse into narrow modes. Note that we do not directly take the difference between the velocity field and the negative score, since this would involve a factor of $(1 - t)$ in the denominator, which becomes problematic as $t \to 1$; $\eta_t$ is defined to safely enforce alignment while avoiding this

singularity. The second term in Eq. (15) corresponds to the entropy of the Gaussian noise **n** injected at the flow terminal, enhancing stochasticity and encouraging diverse exploration. Together, these two terms promote trajectory diversity and maintain the expressiveness of continuous normalizing flows in modeling complex and multi-modal action distributions (see Fig. 2).

Importantly, unlike previous entropy regularizers that require computing log-likelihoods, expensive divergence integration (Chen et al., 2018; Tian et al., 2024) or heuristic noise injection (Frans et al., 2025; Wang et al., 2024), the Brownian regularizer provides a principled yet computationally lightweight alternative.

**Remark** *The Brownian regularizer should not be regarded as a theoretically exact derivation. In particular, while our formulation leverages the relationship between the velocity field and score function under rectified flows, the velocity field in our policy is not obtained via flow matching gradients, and thus does not strictly correspond to the rectified flow dynamics.*

## 5 EXPERIMENTS

We benchmark PolicyFlow against FPO and DPPO because both methods extend PPO to generative policy classes that do not allow explicit likelihood evaluation. These algorithms currently represent the SOTA in applying on-policy RL to expressive, non-Gaussian policy parameterizations. Therefore, comparing to FPO and DPPO is essential for demonstrating the effectiveness of PolicyFlow as a general and principled alternative for training generative policies.

We evaluate these algorithms across the benchmarks in MuJoCo Playground (Zakka et al., 2025) and IsaacLab (Mittal et al., 2023). Using the MultiGoal environment, we test the Brownian regularizer's role in fully leveraging continuous normalizing flow to capture complex, multimodal distributions and avoid mode collapse.

### 5.1 MULTIGOAL TEST

The MultiGoal environment, originally proposed by Haarnoja et al. (2017), is a two-dimensional square workspace with six fixed goal locations, which we use to evaluate how the Brownian regularizer prevents mode collapse and how continuous normalizing flows enable more expressive, multi-modal policies. We modify the dynamics to a second-order system: the state includes position and velocity, and the action is acceleration. Full agent and environment details are provided in Appendix C.2.

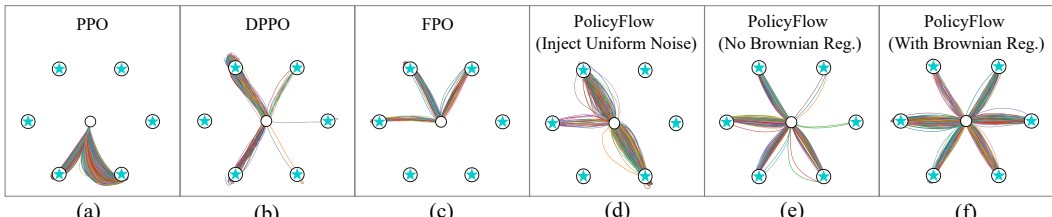

Figure 2: MultiGoal Test (Appendix C.2): sample 1000 trajectories starting at the same original point. (a) PPO with Gaussian entropy regularization ($w_g = 0.001$) covers only a limited set of goals. (b,c) DPPO and FPO collapse to a small number of modes, likely because neither method incorporates any form of entropy regularization. (d) PolicyFlow with uniform noise injection (Ding et al., 2024) (weight 0.05) still suffers from mode collapse, concentrating on only a few modes. (e) PolicyFlow with only Gaussian entropy regularization ($w_g = 0.001$) partially alleviates mode collapse. (f) PolicyFlow with the proposed Brownian regularizer ($w_b = 0.25$) and Gaussian entropy regularization ($w_g = 0.001$) achieves the most diverse and more balanced goal-reaching behaviors.

If the agent starts at the workspace center, all six goals are equidistant and rewards are symmetric, so an optimal policy should reach each goal with roughly equal probability, reflecting the multi-modal nature of the task. As shown in Fig. 2, PPO, which employs a Gaussian policy, can only represent simple distributions and thus struggles to produce trajectories that reach all six goal locations. While

FPO and DPPO utilize generative models capable of expressing more complex distributions, the lack of an effective entropy regularization mechanism prevents the agent from learning a sufficiently diverse set of trajectories. In contrast, PolicyFlow with the Brownian regularizer fully leverages the expressive power of continuous normalizing flows, resulting in more balanced, multi-modal action patterns and a higher coverage of all goals.

## 5.2 MuJoCo Playground and IsaacLab Benchmarks

**MuJoCo Playground benchmarks.** We evaluate PolicyFlow against current state-of-the-art flow-based methods on the MuJoCo Playground benchmarks, including FPO (McAllister et al., 2025) and DPPO (Ren et al., 2024). All these methods are based on the PPO framework, so we also include PPO as a baseline. FPO represents the policy using continuous normalizing flows (CNFs), while DPPO uses diffusion models. The original implementations of FPO and DPPO do not include explicit entropy regularization, which can limit the diversity of the learned policies. As previously shown in the MultiGoal test, PolicyFlow effectively preserves multi-modal behaviors and diverse trajectories. Across the MuJoCo Playground tasks, PolicyFlow achieves performance comparable to or exceeding FPO in most environments, outperforming DPPO, and generally matching or surpassing PPO. Careful examination of the training curves reveals that PolicyFlow often converges faster, indicating higher sample efficiency and effective exploration, complementing the observations on policy diversity from the MultiGoal experiments.

The hyperparameter settings for PolicyFlow are provided in Appendix C.4. To ensure a fair comparison, the hyperparameters for FPO and DPPO follow the tuned configurations from the FPO paper, and PPO uses the default settings recommended by the MuJoCo Playground repository.

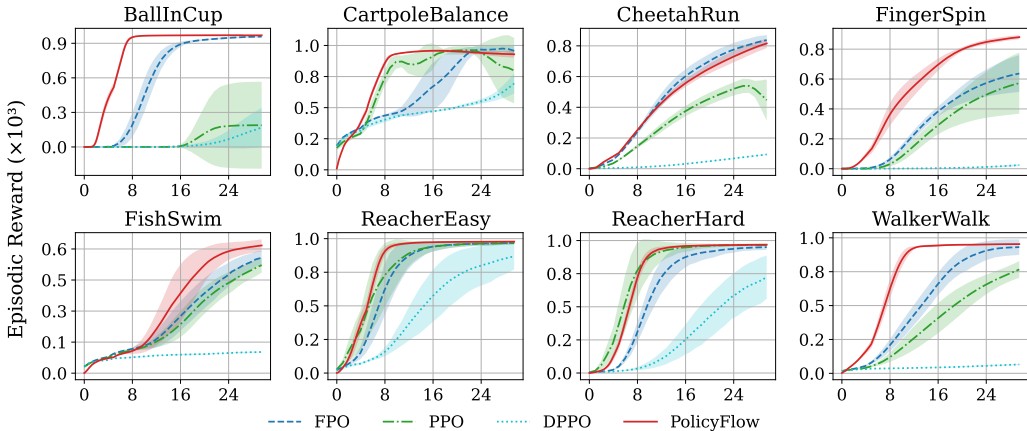

Figure 3: Learning curves on MuJoCo Playground benchmarks. Plots show mean episodic reward with standard error (y-axis) over environment steps (x-axis, total 30M steps), averaged over 5 random seeds.

**IsaacLab benchmarks.** We further evaluate PolicyFlow on the IsaacLab benchmarks, a suite of robotics environments spanning locomotion, manipulation, and navigation. IsaacLab is a recently developed and rapidly growing framework maintained by NVIDIA, designed specifically for large-scale robot learning. Its high simulation fidelity, strong engineering support, and increasing popularity in the robotics community make it an ideal testbed for assessing the performance of RL algorithms. In this benchmark, we compare PolicyFlow only against PPO. Although FPO and DPPO are state-of-the-art generative policy approaches, neither of them includes IsaacLab tasks in their original benchmark suites, and directly adapting their publicly released code to the IsaacLab environment stack requires substantial engineering effort and nontrivial environment re-integration. Therefore, we use the PPO implementation from `RSL-RL` as our baseline, following the official IsaacLab hyperparameter configurations. PolicyFlow uses almost identical hyperparameters to PPO (see Appendix C.3), except for additions required by our entropy regularization mechanism. Full parameter details are provided in Appendix C.3. As shown in Table 1 and Fig. 5 in Appendix C.3, PolicyFlow achieves asymptotic performance that consistently matches or surpasses PPO across all tasks. Since PolicyFlow learns a time-dependent velocity field rather than a direct action mapping, the optimization problem is inherently more complex, which can lead to slower early-stage learning.

Table 1: Terminal training episodic rewards across IsaacLab benchmarks.

| Method | Lift-Cube | Navigation | Open-Drawer | Quadcopter | Anymal-D | G1 | H1 | Go2 |
|---|---|---|---|---|---|---|---|---|
| PPO | $153.1 \pm 3.0$ | $3.5 \pm 0.3$ | $\mathbf{99.8 \pm 1.7}$ | $\mathbf{141.8 \pm 0.5}$ | $24.5 \pm 0.1$ | $25.4 \pm 1.2$ | $\mathbf{29.3 \pm 0.9}$ | $\mathbf{27.9 \pm 0.3}$ |
| PolicyFlow | $\mathbf{154.6 \pm 0.6}$ | $\mathbf{4.2 \pm 0.1}$ | $99.1 \pm 0.7$ | $141.0 \pm 0.09$ | $\mathbf{24.6 \pm 0.2}$ | $\mathbf{30.0 \pm 1.1}$ | $27.3 \pm 0.2$ | $27.4 \pm 0.9$ |
| $p$-value | $0.32$ | $\mathbf{0.0027}$ | $0.41$ | $0.099$ | $0.26$ | $\mathbf{0.00026}$ | $\mathbf{0.0069}$ | $0.33$ |

**Training time per iteration.** We compare the per-iteration training time of PolicyFlow and PPO on IsaacLab environments, which reflects the computational cost of a single training step. As shown in Table 2, when the model parameters are roughly comparable to PPO, PolicyFlow increases per-iteration training time by less than 50% for the first six IsaacLab environments. Even when embedding dimensions are increased up to eightfold, the computational cost remains below twice that of PPO, demonstrating that PolicyFlow is efficient in practice.

Table 2: Per-iteration training time of PPO and PolicyFlow on IsaacLab benchmarks, averaged over 50 iterations on an RTX 5090 GPU.

| Environment | Lift-Cube | Navigation | Open-Drawer | Quadcopter | Anymal-D | G1 | H1 | Go2 |
|---|---|---|---|---|---|---|---|---|
| Embedding Dimensions (Time / Observation) | 64 | 64 | 64 | 64 | 64 | 256 | 512 | 512 |
| PPO (ms) | $43.0 \pm 16.1$ | $36.9 \pm 6.3$ | $81.3 \pm 14.7$ | $37.8 \pm 13.8$ | $41.2 \pm 13.4$ | $66.9 \pm 14.4$ | $63.4 \pm 15.5$ | $63.9 \pm 15.7$ |
| PolicyFlow (ms) | $57.7 \pm 20.8$ | $54.1 \pm 10.1$ | $104.1 \pm 16.5$ | $55.6 \pm 15.3$ | $57.1 \pm 15.7$ | $90.6 \pm 12.3$ | $115.5 \pm 17.3$ | $111.5 \pm 15.1$ |

**Remark** *We do not provide a direct comparison with FPO or DPPO because the implementations of these algorithms in the FPO open-source codebase are based on JAX, whereas PolicyFlow is implemented in PyTorch. Conducting a direct comparison across different deep learning frameworks could lead to unreliable results, so we focus the analysis on PPO, which is implemented in the same framework and provides a fair baseline.*

## 5.3 SENSITIVITY TO CLIPPING RANGE PARAMETER

Our proposed approximation Eq. (13) of the importance ratio introduces an approximation error that is theoretically bounded by the clipping range $\epsilon$, as shown in Appendix A. A smaller $\epsilon$ yields a tighter bound and therefore reduces the approximation error; however, it also limits the effective update step size in policy optimization, which may slow down policy improvement. To empirically verify this trade-off, we conduct a sensitivity analysis in the IsaacLab ANYmal-D environment by evaluating four clipping ranges, $\epsilon \in \{0.1, 0.2, 0.3, 0.4\}$, each with five random seeds. The results shown as Fig. 4a confirm our theoretical insight: smaller clipping ranges lead to lower approximation error but hinder learning progress due to overly conservative updates. Across all IsaacLab benchmarks, we adopt $\epsilon = 0.2$ as the default clipping range, which aligns with the official PPO configuration provided by IsaacLab.

## 5.4 SENSITIVITY TO NETWORK INITIALIZATION AND TIME SAMPLING STRATEGY

We also investigate how different network initialization strategies affect performance. A common choice for MLPs network is the Glorot initialization (Glorot & Bengio, 2010), which samples weights from a Gaussian distribution with an appropriate variance. Alternatively, one may initialize all network parameters to zero. In our study, we compare three initialization schemes in the IsaacLab ANYmal-D environment: (i) **GI**: standard Glorot initialization, (ii) **GI+ZOL**: Glorot initialization with the output layer additionally set to zero, and (iii) **ZI**: full zero initialization for all parameters. As shown in Fig. 4b, the Glorot initialization with a zeroed output layer achieves the best empirical performance. Therefore, this scheme is adopted for initializing our models across all benchmark experiments.

In addition, time sampling is required to estimate the expectation over time $t$ in the objective function Eq. (12). We evaluate several time–sampling strategies in the IsaacLab Navigation environment: (i) **USC**: Uniform sampling from the continuous interval $[0, 1]$; (ii) **USD**: Uniform sampling from the discrete ODE simulation time grid $\{0, 0.05, 0.1, \ldots, 0.95, 1.0\}$; and (iii) **Multi-USD**: sampling multiple time points $t$ for each state–action sample, with each $t$ drawn uniformly from the same discrete grid. The results are shown in Fig. 4c. Overall, these three time–sampling strategies lead to only minor performance differences. Therefore, USD is used as the default choice in all benchmark experiments. Multi-USD is generally not recommended, as it introduces additional computational overhead without clear benefits.

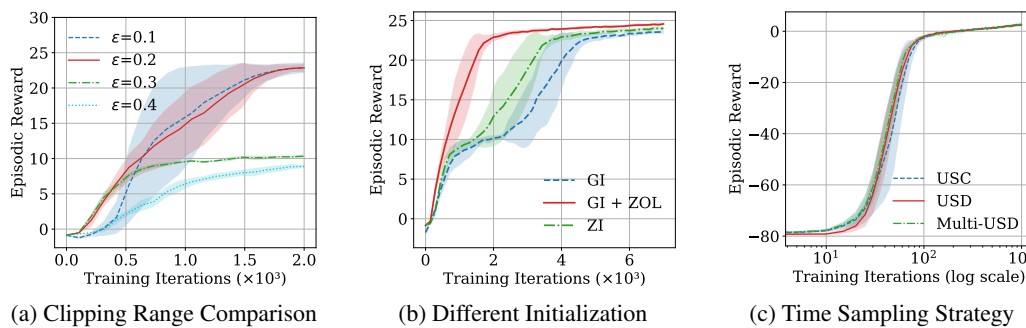

Figure 4: Ablation studies on key components of PolicyFlow.

## 5.5 DIFFERENT CHOICES OF INTERPOLATION PATH

In the preceding sections, we adopted the same interpolation path as rectified flow. However, prior work on flow matching has explored alternative interpolation strategies. Table 4 summarizes two additional choices beyond rectified-flow path. We evaluate these interpolation paths on one IsaacLab locomotion task and the Multi-

Table 3: Terminal training episodic rewards using different interpolation paths.

| Path / Env | ANYmal-D | MultiGoal |
|---|---|---|
| Rectified-Flow Path | $24.6 \pm 0.2$ | $\mathbf{8.79 \pm 0.02}$ |
| Stochastic-Interpolant Path | $\mathbf{24.7 \pm 0.2}$ | $8.22 \pm 0.18$ |
| TrigFlow Path | $24.5 \pm 0.1$ | $8.74 \pm 0.03$ |

Goal test, keeping all agent settings identical except for the interpolation scheme and its corresponding Brownian regularizer. As shown in Table 3, all three paths achieve nearly identical converged episodic rewards on the ANYmal-D locomotion task, whereas TriFlow path and rectified-flow path yield better performance than stochastic-interpolant path on the MultiGoal test. The slightly lower performance of the stochastic-interpolant path may be due to our use of an approximate relationship between the score function and the velocity field along this interpolation, rather than an exact equality. See Appendix B for the details of this approximation.

Table 4: Different interpolation paths for flow matching

| Method / Interpolation Formula | Relationship between Score and Velocity Field |
|---|---|
| Rectified Flow (Liu et al., 2022)
$\mathbf{x}_t = (1-t)\,\mathbf{x}_0 + t\,\mathbf{x}_1, \quad t \in [0,1]$ | $(1-t)\nabla_{\mathbf{x}} \log p_t(\mathbf{x}) = t\,v_t(\mathbf{x}) - \mathbf{x}$ |
| Stochastic Interpolant (Albergo et al., 2023)
$\mathbf{x}_t = (1-t)\mathbf{x}_0 + t\,\mathbf{x}_1 + \sqrt{2t(1-t)}\,\mathbf{h}, t \in [0,1],\ \mathbf{h} \sim \mathcal{N}(\mathbf{0}, \boldsymbol{I})$ | $\left(2(t-\frac{1}{2})^2 + \frac{1}{2}\right)\nabla_{\mathbf{x}} \log p_t(\mathbf{x}) \approx t\,v_t(\mathbf{x}) - \mathbf{x}$ |
| TrigFlow (Lu & Song, 2024)
$\mathbf{x}_t = \cos(t)\,\mathbf{x}_0 + \sin(t)\,\mathbf{x}_1, \quad t \in [0, \frac{\pi}{2}]$ | $\cos(t)\nabla_{\mathbf{x}} \log p_t(\mathbf{x}) = \sin(t)\,v_t(\mathbf{x}) - \cos(t)\,\mathbf{x}$ |

## 6 CONCLUSION AND FUTURE WORKS

In this work, we proposed PolicyFlow, an on-policy reinforcement learning algorithm that integrates continuous normalizing flows with PPO-style optimization. By approximating importance ratios via velocity field variations along interpolation paths, PolicyFlow eliminates the need for costly path-wise backpropagation while maintaining stability and efficiency. In addition, our purposed Brownian regularizer provides a principled yet lightweight way to mitigate mode collapse and encourage diverse exploration. Through extensive experiments on MultiGoal, IsaacLab, and MuJoCo Playground, PolicyFlow consistently matches or outperforms PPO and the SOTA methods FPO and DPPO. In particular, results on MultiGoal showcase PolicyFlow's ability to capture complex multimodal action distributions. Looking forward, PolicyFlow offers a versatile foundation for bridging generative modeling and reinforcement learning. While different interpolation paths show promise in practice, though their formal validation and theoretical implications remain to be explored. Other promising directions include fine-tuning flow-matching policies, extending PolicyFlow to offline RL and broader generative modeling tasks, and exploring its connection to diffusion models by incorporating score-based objectives. Finally, developing a more comprehensive theoretical understanding of PolicyFlow may further inspire algorithmic improvements and strengthen its applicability in real-world decision-making.

## REPRODUCIBILITY STATEMENT

We provide detailed algorithm descriptions in Sec. 4 and Algorithm 1, full hyperparameter settings in Appendix C, and MultiGoal environment configurations in Appendix C.2. Our implementation builds upon the public PPO implementations in `RSL-RL` and `SKRL`. Most results are averaged over multiple random seeds, and our conclusions remain reliable under randomness.

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

## USE OF LARGE LANGUAGE MODELS

Large Language Models (LLMs), specifically ChatGPT-5, were used for two purposes: (1) to aid with grammar checking, language polishing, and ensuring formatting consistency; and (2) for retrieval and discovery tasks, such as finding relevant related work. No content, ideas, or scientific claims were generated or altered by LLMs.

## A  ERROR ANALYSIS OF THE POLICYFLOW OBJECTIVE APPROXIMATION

We start from the flow parameterization of the policy, derive a variational expression for the terminal shift, and introduce an interpolation path approximation. We then rigorously analyze the induced error, showing it to be first-order, and justify the approximation's validity within the PPO framework.

### A.1  VARIATIONAL FORMULA FOR THE TERMINAL SHIFT

Let the two flows, generated from random vectors, satisfy

$$\dot{\varphi}_t(\mathbf{z}; \mathbf{s}) = v_t(\varphi_t(\mathbf{z}; \mathbf{s}); \mathbf{s}), \tag{17}$$

$$\dot{\hat{\varphi}}_t(\mathbf{z}; \mathbf{s}) = \hat{v}_t(\hat{\varphi}_t(\mathbf{z}; \mathbf{s}); \mathbf{s}), \tag{18}$$

with the initial condition $\varphi_0(\mathbf{z}; \mathbf{s}) = \hat{\varphi}_0(\mathbf{z}; \mathbf{s}) = \mathbf{z}$. Set the terminal shift as $\delta_{\varphi_t}(\mathbf{z}; \mathbf{s}) := \varphi_t(\mathbf{z}; \mathbf{s}) - \hat{\varphi}_t(\mathbf{z}; \mathbf{s})$ and the velocity field variation as $\delta_{v_t}(\mathbf{x}; \mathbf{s}) := v_t(\mathbf{x}; \mathbf{s}) - \hat{v}_t(\mathbf{x}; \mathbf{s})$.

Linearizing $v_t$ around the reference trajectory $\hat{\varphi}_t$ yields the variational equation

$$\dot{\delta}_{\varphi_t} = \boldsymbol{J}_t \delta_{\varphi_t} + \delta_{v_t}(\hat{\varphi}_t; \mathbf{s}), \quad \text{with} \quad \delta_{\varphi_0} = \mathbf{0}, \tag{19}$$

where the Jacobian is defined as $\boldsymbol{J}_t := \partial_{\mathbf{x}} \hat{v}_t(\mathbf{x}; \mathbf{s})\big|_{\mathbf{x}=\hat{\varphi}_t}$.

Let $\boldsymbol{\Phi}(1,t)$ be the fundamental matrix of $\dot{\boldsymbol{\Phi}}(\tau,t) = \boldsymbol{J}_\tau \boldsymbol{\Phi}(\tau,t)$ with $\boldsymbol{\Phi}(t,t) = \boldsymbol{I}$. Then, the exact terminal shift is

$$\delta_{\varphi_1}(\mathbf{z}; \mathbf{s}) = \int_0^1 \boldsymbol{\Phi}(1,t) \delta_{v_t}(\hat{\varphi}_t(\mathbf{z}; \mathbf{s}); \mathbf{s}) \mathrm{dt}. \tag{20}$$

### A.2  INTERPOLATION PATH APPROXIMATION AND ERROR ANALYSIS

We approximate the complex integral in Eq. (20) using a simpler linear interpolation path:

$$\mathbf{x}_t := (1-t)\mathbf{z} + t\hat{\varphi}_1(\mathbf{z}; \mathbf{s}), \quad t \in [0,1]. \tag{21}$$

The approximation for the terminal shift is $\tilde{\delta}_{\varphi_1} := \int_0^1 \delta_{v_t}(\mathbf{x}_t; \mathbf{s}) \mathrm{dt}$. The error of this approximation is $E = \delta_{\varphi_1} - \tilde{\delta}_{\varphi_1}$. We decompose this error into two components:

$$E = \underbrace{\int_0^1 (\boldsymbol{\Phi}(1,t) - \boldsymbol{I})\delta_{v_t}(\hat{\varphi}_t)\mathrm{dt}}_{E_1} + \underbrace{\int_0^1 \big(\delta_{v_t}(\hat{\varphi}_t) - \delta_{v_t}(\mathbf{x}_t)\big)\mathrm{dt}}_{E_2}. \tag{22}$$

Assume the following:

(A1) The velocity field $\hat{v}_t$ is $\mathcal{C}^1$ in $\mathbf{x}$ with a uniformly bounded Jacobian, $\|\boldsymbol{J}_t\| \leq L$.

(A2) The velocity variation $\delta_{v_t}$ is uniformly bounded, $\|\delta_{v_t}(\mathbf{x}; \mathbf{s})\| \leq \epsilon$, and is Lipschitz in $\mathbf{x}$ with constant $L_\delta = \mathcal{O}(\epsilon)$ uniformly in $(t, \mathbf{s})$. The assumption on $L_\delta$ is justified as the variation itself stems from a small policy update of size $\epsilon$.

The first error term, $E_1$, arises from approximating the fundamental matrix $\boldsymbol{\Phi}(1, t)$ with the identity matrix $\boldsymbol{I}$. Since $\|\boldsymbol{\Phi}(1, t) - \boldsymbol{I}\| = \mathcal{O}(1)$ and $\|\delta_{v_t}\| = \mathcal{O}(\epsilon)$, the magnitude of this term is:

$$\|E_1\| \leq \int_0^1 \|\boldsymbol{\Phi}(1, t) - \boldsymbol{I}\| \cdot \|\delta_{v_t}(\hat{\varphi}_t)\| \mathrm{dt} = \int_0^1 \mathcal{O}(1) \cdot \mathcal{O}(\epsilon) \mathrm{dt} = \mathcal{O}(\epsilon). \tag{23}$$

The second error term, $E_2$, comes from replacing the true trajectory $\hat{\varphi}_t$ with the linear path $\mathbf{x}_t$. The path deviation $\|\hat{\varphi}_t - \mathbf{x}_t\|$ is generally $\mathcal{O}(1)$. Given the Lipschitz assumption on $\delta_{v_t}$:

$$\|E_2\| \leq \int_0^1 L_\delta \|\hat{\varphi}_t - \mathbf{x}_t\| \mathrm{dt} = \int_0^1 \mathcal{O}(\epsilon) \cdot \mathcal{O}(1) \mathrm{dt} = \mathcal{O}(\epsilon). \tag{24}$$

Since both error components are first-order, the total approximation error is also first-order. We thus correct the conclusion from the main text:

$$\delta_{\varphi_1}(\mathbf{z}; \mathbf{s}) = \int_0^1 \delta_{v_t}(\mathbf{x}_t; \mathbf{s}) \mathrm{dt} + \mathcal{O}(\epsilon). \tag{25}$$

### A.3 First-Order Error of the Likelihood Ratio

Let $\mathbf{n} := \mathbf{a} - \hat{\varphi}_1(\mathbf{z}; \mathbf{s})$, $\mathbf{u} := \delta_{\varphi_1}(\mathbf{z}; \mathbf{s})$, and $\tilde{\mathbf{u}} := \int_0^1 \delta_{v_t}(\mathbf{x}_t; \mathbf{s}) \mathrm{dt}$. From Eq. (25), we have $\mathbf{u} = \tilde{\mathbf{u}} + \mathcal{O}(\epsilon)$.

A Taylor expansion of the Gaussian log-density shows that a first-order error in the mean leads to a first-order error in the log-likelihood:

$$\log p_n(\mathbf{n}; \tilde{\mathbf{u}}, \boldsymbol{\sigma}^2) - \log p_n(\mathbf{n}; \mathbf{u}, \boldsymbol{\sigma}^2) = \mathbf{n}^\top \left(\frac{\tilde{\mathbf{u}} - \mathbf{u}}{\boldsymbol{\sigma}}\right) + \mathcal{O}(\epsilon^2) = \mathbf{n}^\top \left(\frac{\mathcal{O}(\epsilon)}{\boldsymbol{\sigma}}\right) = \mathcal{O}(\epsilon). \tag{26}$$

Using Jensen's inequality, we have

$$\log p_n(\mathbf{n}; \tilde{\mathbf{u}}, \boldsymbol{\sigma}^2) \geq \int_0^1 \log p_n(\mathbf{n}; \delta_{v_t}(\mathbf{x}_t; \mathbf{s}), \boldsymbol{\sigma}^2) \, \mathrm{dt} = \mathbb{E}_{p(t)} \left[\log p_n(\mathbf{n}; \delta_{v_t}(\mathbf{x}_t; \mathbf{s}), \boldsymbol{\sigma}^2)\right]. \tag{27}$$

Thus,

$$\mathbb{E}_{p(t)} \left[\log p_n(\mathbf{n}; \delta_{v_t}(\mathbf{x}_t; \mathbf{s}), \boldsymbol{\sigma}^2)\right] - \log p_n(\mathbf{n}; \mathbf{u}, \boldsymbol{\sigma}^2) = \mathcal{O}(\epsilon). \tag{28}$$

Consequently, the error in the log of the importance ratio is first-order:

$$\left| \mathbb{E}_{p(t)} \left[\log \frac{p_n(\mathbf{n}; \delta_{v_t}(\mathbf{x}_t; \mathbf{s}), \boldsymbol{\sigma}^2)}{p_n(\mathbf{n}; \mathbf{0}, \hat{\boldsymbol{\sigma}}^2)}\right] - \log \frac{p_n(\mathbf{n}; \mathbf{u}, \boldsymbol{\sigma}^2)}{p_n(\mathbf{n}; \mathbf{0}, \hat{\boldsymbol{\sigma}}^2)} \right| = \mathcal{O}(\epsilon). \tag{29}$$

While this is a weak bound of the importance ratio error, it is sufficient to ensure the algorithm's practical effectiveness when using the PPO-style surrogate objective for policy improvement. A first-order approximation error means that for small updates, the gradient computed with the approximate objective is a close match to the gradient from the exact objective. The PPO-style clipping mechanism inherently restricts the update size $\epsilon$, which minimizes the impact of this linear error term and preserves the stability of training. Therefore, the approximation remains a valid and computationally efficient method for optimizing continuous normalizing flow policies.

## B Relationship between Score and Velocity Field for Stochastic Interpolants

Consider the stochastic interpolant used in flow matching:

$$\mathbf{x}_t = t\mathbf{x}_1 + (1 - t)\mathbf{x}_0 + \gamma(t)\mathbf{z}, \qquad \mathbf{x}_0 \sim p_0, \ \mathbf{x}_1 \sim p_1, \ \mathbf{z} \sim \mathcal{N}(0, I). \tag{30}$$

Albergo et al. (2023) derive the relationship between the score function $\nabla_{\mathbf{x}} \log p_t(\mathbf{x}_t)$ along the interpolation path and the velocity field $v_t(\mathbf{x}_t)$ used in conditional flow matching (see Eq. (2.27) in their work):

$$v_t(\mathbf{x}_t) = \bar{v}_t(\mathbf{x}_t) - \dot{\gamma}(t)\,\gamma(t)\,\nabla_{\mathbf{x}} \log p_t(\mathbf{x}_t), \tag{31}$$

where

$$\bar{v}_t(\mathbf{x}_t) = \mathbb{E}_t[\mathbf{x}_1 - \mathbf{x}_0 \mid t\mathbf{x}_1 + (1-t)\mathbf{x}_0 + \gamma(t)\mathbf{z} = \mathbf{x}_t]. \tag{32}$$

In general, obtaining a closed-form expression for $\bar{v}_t(\mathbf{x}_t)$ is intractable, which prevents an exact analytic relationship between the score function and the velocity field under a stochastic interpolant. However, when $\gamma(t) = 0$, the interpolant reduces to the deterministic path of rectified flows, for which the relationship becomes explicit. This motivates approximating $\bar{v}_t(\mathbf{x}_t)$ using the deterministic identity:

$$\bar{v}_t(\mathbf{x}_t) \approx \frac{(1-t)\,\nabla_{\mathbf{x}} \log p_t(\mathbf{x}_t) + \mathbf{x}_t}{t}. \tag{33}$$

Substituting this approximation into the expression for $v_t(\mathbf{x}_t)$ gives an approximate relationship between the stochastic velocity field and the score:

$$v_t(\mathbf{x}_t) \approx \frac{(1-t)\,\nabla_{\mathbf{x}} \log p_t(\mathbf{x}_t) + \mathbf{x}_t}{t} - \dot{\gamma}(t)\,\gamma(t)\,\nabla_{\mathbf{x}} \log p_t(\mathbf{x}_t). \tag{34}$$

Finally, choosing the commonly used stochasticity schedule

$$\gamma(t) = \sqrt{2(1-t)t}, \tag{35}$$

yields the expressions summarized in Table 4.

## C EXPERIMENTAL DETAILS

Our algorithm builds upon the open-source frameworks SKRL (Serrano-Muñoz et al., 2023) and RSL-RL (Schwarke et al., 2025). Specifically, we inherit the implementation of the flexible replay buffer from SKRL and integrate it with the framework of the PPO implementation provided by RSL-RL.

### C.1 MODEL ARCHITECTURE

The model used in PolicyFlow is based on a flow network that maps noise inputs to actions, conditioned on state or other context. The model comprises four main components:

**Flow Network (MLP)**: A multi-layer perceptron that predicts the velocity with noised actions and time and observation embedding as inputs.

**Timestep Embedding (FourierEmbedding)**: Uses a fixed set of frequencies to encode the scalar noise / time step into a high-dimensional representation. The embedding is computed as

$$\mathbf{t}_{\text{emb}} = \text{MLP}([\cos(2\pi f_i t), \sin(2\pi f_i t)]_{i=1}^{d/2})$$

which allows the model to better capture temporal dependencies.

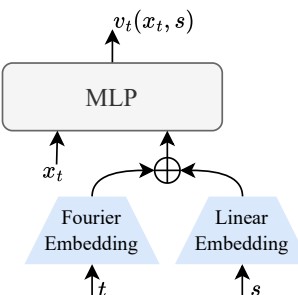

**Observation Embedding (LinearLayer)**: A linear layer that embeds observation vectors into a fixed-dimensional space, which is added to the timestep embedding to modulate the flow network outputs.

**Learnable Variance**: It's the same as that in Gaussian policy parametrization in PPO.

### C.2 MULTIGOAL SETUPS

The MultiGoal environment is a two-dimensional square workspace designed to evaluate the ability of agents to learn multimodal and balanced goal-reaching behaviors. Six fixed goals are placed evenly on a circle centered at the origin, with equal distance to the starting position. At the beginning

of each episode, the agent is randomly initialized within the workspace and must learn to reach the nearest goal as efficiently as possible.

The environment is modeled as a second-order system: the state consists of both position and velocity, and the action corresponds to a 2D acceleration vector. The reward is composed of two components: a distance-based term encouraging the agent to approach the nearest goal, and an action penalty discouraging excessive control inputs. By combining these terms, the environment provides a consistent evaluation of both goal-reaching accuracy and control efficiency.

The Table 5 and Table 6 summarize the main environment configuration parameters and the definitions of state, observation, and reward functions.

The agent configurations for the MultiGoal task are summarized in Table 7 and Table 8. Both PolicyFlow and PPO share common hyperparameters such as learning rate, discount factor, GAE parameter, and clipping settings. Compared to PPO, PolicyFlow introduces an additional term, Brownian regularization loss, while PPO employs a standard Gaussian entropy regularizer.

Table 5: MultiGoal environment configuration

| Parameter | Value |
|---|---|
| Maximum episode length | 50 |
| Action clipping range | [-5.0, 5.0] |
| Integration timestep ($\delta t$) | 0.1 |
| Check wall collisions | True |
| Workspace $x$-limits | [-8.0, 8.0] |
| Workspace $y$-limits | [-8.0, 8.0] |
| Number of goals | 6 |
| Goal radius | 5.0 |
| Goal center | [0.0, 0.0] |
| Distance reward scale | 1.0 |
| Action penalty scale | 0.01 |

Table 6: MultiGoal Environment: State/Observation and Reward Functions

| Component | Description / Formula |
|---|---|
| **Observation / State** | |
| State vector $\mathbf{s}$ | $\mathbf{s} = [x, y, v_x, v_y]$ |
| Actor / Critic observation | $\mathbf{o}_{\text{actor}} = \mathbf{o}_{\text{critic}} = \mathbf{s}$ |
| **Action** | |
| Action $\mathbf{a}$ | 2D acceleration: $\mathbf{a} = [a_x, a_y]$ (clipped to $[-5, 5]$) |
| **Reward Functions** | |
| Distance reward | $r_{\text{distance}} = \exp(-0.3\, d_{\text{min}}^2) + \exp(-0.1\, d_{\text{min}}^2)$, where $d_{\text{min}} = \min_i \|\mathbf{p} - \mathbf{g}_i\|$ is the distance to the nearest goal $\mathbf{g}_i$ |
| Action penalty | $r_{\text{action}} = -0.01 \|\mathbf{a}\|^2$ |
| Total reward | $r_{\text{total}} = r_{\text{distance}} + r_{\text{action}}$ |

Table 7: PPO and PolicyFlow hyperparameter configurations for MultiGoal

| Parameter | PolicyFlow | PPO |
|---|---|---|
| Desired KL divergence | 0.01 | 0.01 |
| Learning rate | $2 \times 10^{-4}$ | $2 \times 10^{-4}$ |
| Discount factor $\gamma$ | 0.99 | 0.99 |
| GAE $\lambda$ | 0.95 | 0.95 |
| Time-limit bootstrap | True | True |
| Mini-batches | 4 | 4 |
| Learning epochs | 5 | 5 |
| Gaussian entropy loss scale | 0.001 | 0.001 |
| Brownian regularization loss scale | 0.25 | — |
| Ratio clip $\epsilon$ | 0.2 | 0.2 |
| Clip predicted values | True | True |
| Value clip | 0.2 | 0.2 |
| Value loss scale | 1.0 | 1.0 |
| Timesteps of second-order Runge–Kutta integration | 12 | |
| Actor hidden layers | [256, 128, 64] | [256, 128, 64] |
| Critic hidden layers | [256, 128, 64] | [256, 128, 64] |
| Actor/Critic activation | Mish | Mish |
| Number of parallel environments | 1024 | 1024 |
| Rollouts per environment | 24 | 24 |

Table 8: Hyperparameter settings for Flow Matching Policy Optimization (FPO) on the MultiGoal environment.

| Parameter | Value |
|---|---|
| Timesteps of Euler integration | 12 |
| Time embedding dimension | 8 |
| Samples number per action | 8 |
| Average losses before exponentiation | True |
| Use discretized timesteps in training | True |
| Clipping epsilon for ratio update | 0.05 |
| Training batch size | 1024 |
| Discount factor $\gamma$ | 0.99 |
| Maximum steps per episode | 50 |
| Learning rate | $1 \times 10^{-4}$ |
| Number of parallel environments | 4096 |
| Mini-batches per update | 32 |
| Total environment steps | $1.2 \times 10^{8}$ |
| Learning epochs | 16 |
| GAE parameter $\lambda$ | 0.95 |
| Value loss scale | 0.25 |
| Actor hidden layers | [256, 128, 64] |
| Critic hidden layers | [256, 128, 64] |
| Actor/Critic activation | Mish |

## C.3 ISAACLAB

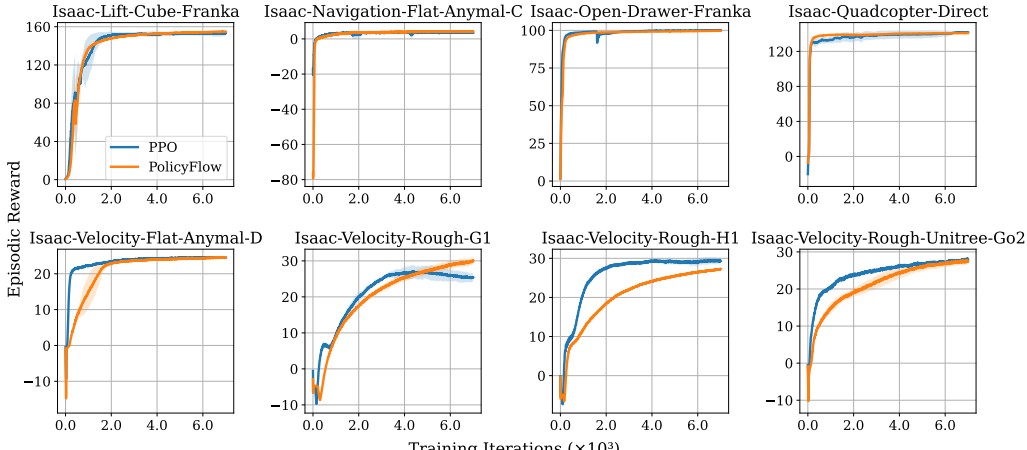

Figure 5: Learning curves (PolicyFlow v.s. PPO) on IsaacLab benchmarks. Plots show mean episodic reward with standard error (y-axis) over training iterations (x-axis), averaged over 5 random seeds.

Table 9: Common hyperparameter settings for PPO and PolicyFlow used in the all IsaacLab benchmarks

| Parameter | PPO | PolicyFlow |
|---|---|---|
| Timesteps of second-order Runge–Kutta integration | – | 12 |
| Number of parallel environments | 4096 | 4096 |
| Learning rate adaptive | True | True |
| Desired KL target | 0.01 | 0.01 |
| GAE parameter $\lambda$ | 0.95 | 0.95 |
| Time-limit bootstrap | True | True |
| Mini-batches per update | 4 | 4 |
| Learning epochs | 5 | 5 |
| Ratio clip $\epsilon$ | 0.2 | 0.2 |
| Clip predicted values | True | True |
| Value clip | 0.2 | 0.2 |
| Value loss scale | 1.0 | 1.0 |
| Gradient norm clip | 1.0 | 1.0 |
| Actor/Critic activation | ELU | ELU |

Table 10: Variable hyperparameter settings for PPO and PolicyFlow on three IsaacLab benchmarks.

| Parameter | Lift-Cube-Franka | | Navigation-ANYmal-C | | Quadcopter-Direct | |
|---|---|---|---|---|---|---|
| | PPO | PolicyFlow | PPO | PolicyFlow | PPO | PolicyFlow |
| Actor/Critic hidden layers | [256,128,64] | [256,128,64] | [128,128] | [128,128] | [64,64] | [128,128] |
| Initial learning rate | $1 \times 10^{-4}$ | $1 \times 10^{-4}$ | $1 \times 10^{-3}$ | $1 \times 10^{-3}$ | $5 \times 10^{-4}$ | $5 \times 10^{-4}$ |
| Discount factor $\gamma$ | 0.98 | 0.98 | 0.99 | 0.99 | 0.99 | 0.99 |
| Gaussian entropy coefficient $w_g$ | 0.006 | 0.004 | 0.005 | 0.0025 | 0.01 | 0.005 |
| Brownian regularizer coefficient $w_b$ | – | 0.002 | – | 0.0025 | – | 0.005 |
| Time/Observation embedding dim | – | 64 | – | 64 | – | 64 |
| Rollouts per environment | 24 | 24 | 8 | 8 | 24 | 24 |

Table 11: Variable hyperparameter settings for PPO and PolicyFlow on three IsaacLab benchmarks.

| Parameter | Flat-ANYmal-D | | Rough-H1 | | Rough-Unitree-Go2 | |
|---|---|---|---|---|---|---|
| | PPO | PolicyFlow | PPO | PolicyFlow | PPO | PolicyFlow |
| Actor/Critic hidden layers | [128,128,128] | [128,128,128] | [512,256,128] | [512,256,128] | [512,256,128] | [512,256,128] |
| Initial learning rate | $1 \times 10^{-3}$ | $1 \times 10^{-3}$ | $1 \times 10^{-3}$ | $1 \times 10^{-3}$ | $1 \times 10^{-3}$ | $1 \times 10^{-3}$ |
| Discount factor $\gamma$ | 0.99 | 0.99 | 0.99 | 0.99 | 0.99 | 0.99 |
| Gaussian entropy coefficient $w_g$ | 0.005 | 0.0025 | 0.01 | 0.005 | 0.01 | 0.008 |
| Brownian regularizer coefficient $w_b$ | – | 0.0025 | – | 0.005 | – | 0.002 |
| Time/Observation embedding dim | – | 64 | – | 512 | – | 512 |
| Rollouts per environment | 24 | 24 | 24 | 24 | 24 | 24 |

Table 12: Variable hyperparameter settings for PPO and PolicyFlow on two IsaacLab benchmarks.

| Parameter | Open-Drawer-Franka | | Rough-G1 | |
|---|---|---|---|---|
| | PPO | PolicyFlow | PPO | PolicyFlow |
| Actor/Critic hidden layers | [256,128,64] | [256,128,64] | [512,256,128] | [512,256,128] |
| Initial learning rate | $5 \times 10^{-4}$ | $5 \times 10^{-4}$ | $1 \times 10^{-3}$ | $1 \times 10^{-3}$ |
| Discount factor $\gamma$ | 0.99 | 0.99 | 0.99 | 0.99 |
| Gaussian entropy coefficient $w_g$ | 0.001 | 0.0008 | 0.008 | 0.002 |
| Brownian regularizer coefficient $w_b$ | – | 0.0002 | – | 0.006 |
| Time/Observation embedding dim | – | 64 | – | 256 |
| Rollouts per environment | 96 | 96 | 24 | 24 |

## C.4 MuJoCo Playground

Table 13: Common hyperparameters for PolicyFlow across MuJoCo Playground benchmarks.

| Parameter | Value |
|---|---|
| Discount factor $\gamma$ | 0.995 |
| GAE parameter $\lambda$ | 0.95 |
| Time-limit bootstrap | True |
| Learning rate adaptive | False |
| Mini-batches per update | 4 |
| Learning epochs | 16 |
| Clip predicted values | False |
| Value loss scale | 0.25 |
| Timesteps of second-order Runge–Kutta integration | 12 |
| Actor hidden layers | [32, 32, 32, 32] |
| Critic hidden layers | [256, 256, 256, 256, 256] |
| Actor/Critic activation | SiLU |
| Number of parallel environments | 1024 |
| Rollouts per environment | 24 |

Table 14: Variable hyperparameters for PolicyFlow on MuJoCo Playground benchmarks (1/2).

| Parameter | WalkerWalk | FingerSpin | BallInCup | CartpoleBalance | FishSwim |
|---|---|---|---|---|---|
| Learning rate | $1 \times 10^{-3}$ | $5 \times 10^{-5}$ | $3 \times 10^{-4}$ | $5 \times 10^{-5}$ | $3 \times 10^{-4}$ |
| Gaussian entropy coefficient $w_g$ | 0.001 | 0.01 | 0.002 | 0.001 | 0.008 |
| Brownian regularizer coefficient $w_b$ | 0.001 | 0.001 | 0.001 | 0.0001 | 0.002 |
| Ratio clip $\epsilon$ | 0.2 | 0.2 | 0.2 | 0.01 | 0.2 |
| Gradient norm clip | 1.0 | 10.0 | 1.0 | 0.5 | 5.0 |

Table 15: Variable hyperparameters for PolicyFlow on MuJoCo Playground benchmarks (2/2).

| Parameter | FingerTurnHard | FingerTurnEasy | CheetahRun | ReacherEasy | ReacherHard |
|---|---|---|---|---|---|
| Learning rate | $4 \times 10^{-4}$ | $3 \times 10^{-4}$ | $3 \times 10^{-4}$ | $3 \times 10^{-4}$ | $3 \times 10^{-4}$ |
| Gaussian entropy coefficient $w_g$ | 0.006 | 0.002 | 0.002 | 0.002 | 0.002 |
| Brownian regularizer coefficient $w_b$ | 0.001 | 0.001 | 0.001 | 0.001 | 0.001 |
| Ratio clip $\epsilon$ | 0.2 | 0.05 | 0.05 | 0.05 | 0.05 |
| Gradient norm clip | -1.0 | 10.0 | 10.0 | 5.0 | 5.0 |

