# OpenReview forum: "PolicyFlow: Policy Optimization with Continuous Normalizing Flow in Reinforcement Learning"
_ICLR.cc/2026/Conference — ICLR 2026 Poster_

### Official Review · Reviewer_P1yA · 2025-10-29

**Soundness:** 3
**Presentation:** 3
**Contribution:** 3
**Rating:** 4
**Confidence:** 5

**Summary:**

This paper proposes a novel on-policy algorithm named PolicyFlow. Its core contributions are:

1) Efficient importance ratio approximation: Instead of computing the full flow path during training updates, PolicyFlow approximates importance ratios using velocity field variations along a simple "interpolation path". This significantly reduces computational overhead.
2) Brownian entropy regularizer: To prevent potential mode collapse and encourage diverse behaviors , the authors introduce a lightweight entropy regularizer inspired by Brownian motion. It implicitly increases policy entropy by shaping the velocity field , avoiding the significant cost of explicit entropy or log-likelihood computation.

The authors conducts experiments in diverse environments, including MultiGoal, IsaacLab, and MuJoCo Playground. Particularly on the MultiGoal task, PolicyFlow demonstrates its ability to capture diverse multimodal action distributions.

**Strengths:**

1) The core contribution of this paper is a novel and efficient importance ratio approximation method. By evaluating velocity field differences along a simple interpolation path, it avoids the significant overhead of solving ODEs and backpropagation during PPO updates, making the application of CNF policies computationally feasible. Additionally, inspired by physical processes, the paper proposes a novel Brownian regularizer, which provides a lightweight means of preventing mode collapse by directly acting on the velocity field, thereby avoiding expensive entropy calculations.
2) The method of this paper performs well on MultiGoal (Figure 1) and demonstrates high computational efficiency.

**Weaknesses:**

1) The baselines are limited, comparing only against PPO and FPO. Methods like GenPO and DPPO are missing. Furthermore, the method does not show an advantage in half of the tasks in the IsaacLab benchmarks.

2) There are too many hyperparameters, making reproduction difficult. Beyond standard PPO hyperparameters, it also involves the sampling strategy for $t$, noise variance learning, time or observation embedding dimensions, integration steps, and $w_b$, $w_g$, among others. The paper provides limited discussion on the sensitivity to these hyperparameters and the default strategies used.

3) No code available.

**Questions:**

1) Table 5 shows that using different interpolation paths leads to drastically different performance on the MultiGoal task. This suggests that the choice of interpolation path is another critical design decision. Why does the paper default to the Rectified-Flow path? What advantages does it have, in theory or practice, compared to other paths?

---

> ### Author Response · Authors · 2025-11-22
>
> > Weakness 1.1: The baselines are limited, comparing only against PPO and FPO. Methods like GenPO and DPPO are missing.
>
> We thank the reviewer for this comment. In the revised paper, **we have added DPPO as a baseline and included comparisons on the MuJoCo Playground benchmark (Fig. 3)**.
>
> Regarding other baselines:
> - GenPO was not included because its code is not publicly available. Using only the hyperparameters reported in their paper, we find that PPO already outperforms the results reported for GenPO on IsaacLab. Since we cannot determine whether this discrepancy arises from differences in IsaacLab versions or implementation details not described in the paper, we did not include GenPO as a baseline.
> -  We evaluated FPO and DPPO only on the MuJoCo Playground because both methods require non-trivial architectural and training-loop modifications that are incompatible with IsaacLab’s current PyTorch-based pipeline: FPO is implemented in JAX, and its training loop, data pipeline, and vectorization mechanism heavily rely on JAX primitives. Porting FPO to PyTorch/IsaacLab would essentially require a full reimplementation; DPPO, while implemented in PyTorch, is designed specifically for fine-tuning diffusion models and does not provide a modular policy interface compatible with IsaacLab’s rollouts.
>
> Adapting DPPO and FPO to IsaacLab would require substantial refactoring. We have already started implementing a compatible version of DPPO and FPO within the IsaacLab framework. If we obtain reliable results, we will include them in the final version of the paper.
>
> ---
>
>
> > Weakness 1.2: Furthermore, the method does not show an advantage in half of the tasks in the IsaacLab benchmarks.
>
> We thank the reviewer for this comment. PolicyFlow is designed as an extension of PPO to enable training continuous normalizing flow (CNF)-based policies, which cannot be handled by standard PPO due to the lack of an explicit probability density function. In tasks where the optimal policy is nearly deterministic, such as many of the IsaacLab benchmarks, a Gaussian policy already provides an excellent approximation. As CNFs strictly generalize Gaussian policies, PolicyFlow naturally recovers similar performance in these tasks, which explains why no clear advantage is observed in roughly half of the benchmarks.
>
> **The main strength of PolicyFlow lies in tasks where multimodal actions or richer exploration are required, such as our MultiGoal (Fig. 2) and newly added PointMaze (Fig. 1) experiments.** In these environments, CNF policies enable the agent to represent complex and multimodal action distributions, resulting in more diverse exploration and improved coverage of possible behaviors. These benefits are less pronounced in standard continuous-control benchmarks with nearly deterministic optimal policies, which is consistent with our empirical observations.
>
> Importantly, in the robotics and visuomotor learning community, **flow-based or generative policies are most commonly trained using offline imitation-learning methods, such as denoising score matching or flow matching**. While effective, these approaches have reached clear performance bottlenecks, especially in complex, **contact-rich, or multimodal tasks**. There is now growing interest in **using reinforcement learning to train these expressive policies, which provides the motivation and practical relevance for PolicyFlow**. Our method offers a principled on-policy RL framework to optimize CNF-based policies, potentially overcoming the limitations of offline training.
>
> This is supported by recent works demonstrating that generative policies can achieve superior performance in challenging robotic manipulation and visuomotor tasks [1–4]. Thus, even if gains are not evident in simpler continuous-control benchmarks, **PolicyFlow provides a potential approach for enabling RL to exploit the expressive power of generative policies, which is a promising direction in modern robotics**.
>
> **Reference**:
>
> [1] Lei, Kun, Huanyu Li, Dongjie Yu, Zhenyu Wei, Lingxiao Guo, Zhennan Jiang, Ziyu Wang, Shiyu Liang, and Huazhe Xu. "RL-100: Performant Robotic Manipulation with Real-World Reinforcement Learning." arXiv preprint arXiv:2510.14830 (2025).
>
> [2] Intelligence, Physical, Kevin Black, Noah Brown, James Darpinian, Karan Dhabalia, Danny Driess, Adnan Esmail et al. "$\pi_ {0.5} $: a Vision-Language-Action Model with Open-World Generalization." arXiv preprint arXiv:2504.16054 (2025).
>
> [3] Gao, Dechen, Boqi Zhao, Andrew Lee, Ian Chuang, Hanchu Zhou, Hang Wang, Zhe Zhao, Junshan Zhang, and Iman Soltani. "Vita: Vision-to-action flow matching policy." arXiv preprint arXiv:2507.13231 (2025).
>
> [4] Chi, Cheng, Zhenjia Xu, Siyuan Feng, Eric Cousineau, Yilun Du, Benjamin Burchfiel, Russ Tedrake, and Shuran Song. "Diffusion policy: Visuomotor policy learning via action diffusion." The International Journal of Robotics Research 44, no. 10-11 (2025): 1684-1704.
>
> ---

---

> ### Author Response · Authors · 2025-11-22
>
> > Weakness 2: There are too many hyperparameters, making reproduction difficult. Beyond standard PPO hyperparameters, it also involves the sampling strategy for 𝑡, noise variance learning, time or observation embedding dimensions, integration steps, and 𝑤𝑏, 𝑤𝑔, among others. The paper provides limited discussion on the sensitivity to these hyperparameters and the default strategies used.
>
> We thank the reviewer for highlighting the concern regarding hyperparameters. To demonstrate the effect of the Brownian regularizer $w_b$, we conducted experiments on MultiGoal and PointMaze. Our empirical findings indicate that starting from a small value (e.g., 0.001) and gradually increasing $w_b$ allows the agent to explore larger regions without negatively affecting episodic rewards, thereby leveraging the CNF’s ability to model complex distributions and learn more diverse behaviors.
>
> In addition, **we have added an ablation study on model initialization in Sec. 5.4**,  illustrating how different initialization strategies influence the algorithm. **We also investigate various time sampling strategies to assess their impact on training in Sec. 5.4**.
>
> Since our proposed approximation method introduces some error that depends on the PPO clipping range $\epsilon$, **we further included experiments comparing different $\epsilon$ values in Sec. 5.3**, providing guidance for selecting this parameter in practice.
>
> Finally, we would like to emphasize that PolicyFlow is not difficult to reproduce. From an implementation perspective, the method primarily involves a **plug-in replacement of PPO’s importance ratio approximation**, so the overall code structure and training pipeline remain very similar to standard PPO. We acknowledge that our original manuscript may not have made this point sufficiently explicit, and we have clarified it in the revised version.
>
> ---
>
> > Weakness 3: No code available.
>
> We thank the reviewer for this comment. We fully recognize that releasing code is **crucial for understanding the implementation details of the algorithm and for ensuring reproducibility of the reported results**. We are currently preparing the public release of our code, including a complete README and the removal of all identifiable information.
>
> Our implementation builds upon the PPO frameworks in RSL-RL and SKRL, enabling seamless integration with the widely adopted IsaacLab environment. In addition, our codebase also supports MuJoCo Playground and Gym environments through lightweight environment wrappers.
>
> For the flow model, we adopt a **modular architecture inspired by CleanDiffuser**(https://github.com/CleanDiffuserTeam/CleanDiffuser), which improves clarity, extensibility, and ease of modification for future research.
>
> We plan to release the code together with the final version of the paper and are committed to ensuring that the implementation is clean, well-documented, and easy for the community to build upon. (We now have a pre-release code available https://github.com/PolicyFlow2026/PolicyFlow)

---

> ### Author Response · Authors · 2025-11-22
>
> > Q1: Table 5 shows that using different interpolation paths leads to drastically different performance on the MultiGoal task. This suggests that the choice of interpolation path is another critical design decision. Why does the paper default to the Rectified-Flow path? What advantages does it have, in theory or practice, compared to other paths?
>
> We thank the reviewer for this careful observation. In comparative tests on the MultiGoal environment, the Stochastic-Interpolant Path performs slightly worse. One contributing factor is that, for this path, **the relationship between the score function in the entropy regularization and the velocity field is only approximate, whereas for the Rectified-Flow Path and TrigFlow Path, this relationship holds exactly**. This approximation arises from Equation 2.27 in Albergo et al. [1], and we have added a detailed derivation of this approximation in Appendix B.
>
> Regarding the choice of interpolation path, Rectified-Flow Path is commonly preferred in flow matching. **Using the Rectified-Flow algorithm minimizes the curvature of the ODE trajectory, which allows larger step sizes during ODE simulation and reduces inference time**. This motivated our choice of this path. We also compare with TrigFlow [2] because their work demonstrates that this path unifies the straightening techniques in EDM [3] and flow matching [4], and performs well in one-step inference.
>
> **Reference**:
>
> [1] Albergo, Michael S., Nicholas M. Boffi, and Eric Vanden-Eijnden. "Stochastic interpolants: A unifying framework for flows and diffusions, 2023." URL https://arxiv. org/abs/2303.08797 3 (2023).
>
> [2] Lu, Cheng, and Yang Song. "Simplifying, stabilizing and scaling continuous-time consistency models." arXiv preprint arXiv:2410.11081 (2024).
>
> [3] Karras, Tero, Miika Aittala, Timo Aila, and Samuli Laine. "Elucidating the design space of diffusion-based generative models." Advances in neural information processing systems 35 (2022): 26565-26577.
>
> [4] Lipman, Yaron, Ricky TQ Chen, Heli Ben-Hamu, Maximilian Nickel, and Matt Le. "Flow matching for generative modeling." arXiv preprint arXiv:2210.02747 (2022).

---

> ### Author Response · Authors · 2025-11-22
>
> We thank the reviewer for the thorough and insightful assessment. We appreciate the positive recognition of our key contributions, especially the efficient importance ratio approximation and the Brownian regularizer, and the constructive feedback regarding baseline coverage, hyperparameter sensitivity, and code availability. These comments are extremely helpful for improving the clarity and completeness of the paper.

---

### Official Review · Reviewer_gsX4 · 2025-10-31

**Soundness:** 3
**Presentation:** 2
**Contribution:** 3
**Rating:** 4
**Confidence:** 3

**Summary:**

The paper introduces PolicyFlow, an on-policy RL method that uses continuous normalizing flows for more expressive policies. It avoids expensive backpropagation through flow trajectories by using an interpolation-based importance ratio. A Brownian regularizer is added to encourage exploration. Experiments on MuJoCo, IsaacLab, and MultiGoal show stable and competitive results compared to PPO and FPO.

**Strengths:**

The paper proposes an interesting and original idea — combining continuous normalizing flows with on-policy policy optimization in a practical way.
The paper is clearly written, with intuitive explanations and helpful figures that make the method easy to understand.

Experiments cover multiple benchmarks (MuJoCo, IsaacLab, MultiGoal) and show consistent improvements over PPO and FPO.

Overall, the work is technically solid and provides a promising direction for expressive yet stable flow-based policy learning.

**Weaknesses:**

Methodological Weaknesses:
1. The proposed interpolation-based estimation of importance ratios is only heuristic; the paper does not quantify the bias introduced or establish convergence guarantees. Providing analytical error bounds or controlled experiments comparing with exact estimators would strengthen credibility.
2. The Brownian regularizer, while novel, lacks clear motivation and comparison with existing entropy regularizers (e.g., Haarnoja et al., 2018; Chao et al., 2024). Its empirical benefit is only qualitatively shown; a quantitative ablation isolating its contribution is necessary.


Experiments Weaknesses:
1. The experimental evaluation is also fragmented—PolicyFlow is compared with FPO on MuJoCo and PPO on IsaacLab, preventing a unified cross-method assessment.
2. Including all three under identical settings would clarify relative efficiency and expressivity. Furthermore, results are averaged over five seeds without significance testing
3. The MuJoCo Playground evaluation covers only a moderate subset of tasks (e.g., BallInCup, CheetahRun, FingerTurnHard, ReacherHard) but omits the most challenging or high-dimensional environments such as Humanoid or Walker. This limits claims of scalability to complex continuous-control domains.

Minor Issues:
1. some notations is not well defined: Sec. 3. $s,a$ is not defined before using. $p_{\pi}(s)$ is never defined. $\pi^*$ is mentioned but not clearly defined.

**Questions:**

Can you clarify whether the interpolation-based importance ratio introduces any noticeable bias? It would help to know if you’ve compared it against an exact or less-approximated version.

---

> ### Author Response · Authors · 2025-11-22
>
> We thank the reviewer for the constructive and thoughtful feedback. We appreciate the recognition of our main ideas and experimental results, and we value the comments regarding the importance ratio bias, the Brownian regularizer, and the experimental setup. These suggestions are very helpful. We will address the concerns in the rebuttal with additional analysis and clarifications, and we are running further experiments that we will include in the final version.
>
> ---
>
> > Weakness 1: The proposed interpolation-based estimation of importance ratios is only heuristic; the paper does not quantify the bias introduced or establish convergence guarantees. Providing analytical error bounds or controlled experiments comparing with exact estimators would strengthen credibility.
>
> We thank the reviewer for this insightful comment. Indeed, diffusion/flow-based models do not explicitly model the probability density function, and computing the exact likelihood for such models is computationally equivalent to solving the corresponding SDE or its Fokker-Planck PDE, which is highly challenging.
>
> Our proposed interpolation-based estimation of importance ratios is motivated by the intuitive idea that the change at the flow endpoint is approximately equal to the accumulated change of the velocity along the interpolation path. Based on this, we replace the Gaussian policy in PPO with a flow model and provide an approximate importance ratio wherever it is required.
>
> While we do not provide a fully rigorous theoretical justification for this approximation, **we have analyzed the induced error in the Appendix A and, in relation to the PPO clipping range $\epsilon$, provided an error bound (see Line 216)**. This approximation is indeed biased, but the bias can be controlled by the PPO clip parameter $\epsilon$. **We have further conducted an ablation study on this parameter to demonstrate its effect on training (Sec. 5.3)**.
>
> Both FPO and our method rely on approximating the importance ratio, which introduces an asymmetric bias, motivating our comparison with FPO as a baseline. Moreover, similar approximations appear in recent work such as DiffusionNFT, suggesting a potentially unifying theoretical explanation that could be explored in future work.
>
> ---
>
> > Weakness 2: The Brownian regularizer, while novel, lacks clear motivation and comparison with existing entropy regularizers (e.g., Haarnoja et al., 2018; Chao et al., 2024). Its empirical benefit is only qualitatively shown; a quantitative ablation isolating its contribution is necessary.
>
> We thank the reviewer for this valuable comment. To better address this concern, **we have added comparative experiments in the MultiGoal (Fig. 2) and PointMaze (Fig. 1) environments**, where we evaluate our Brownian regularizer against heuristic noise injection methods [1] and standard regularization for Gaussian. These experiments provide quantitative evidence of its contribution.
>
> Regarding the motivation and comparison with existing entropy regularizers, we would like to clarify the following. For diffusion/flow-based models that implicitly model distributions via SDEs or ODEs, computing entropy is intractable. As a result, there is currently no straightforward method to directly apply entropy regularization to flow-based policies.
>
> In contrast, the entropy regularizer in Haarnoja et al. (2018) is applicable to Gaussian policies or other models with explicit PDFs, where the entropy can be computed directly; such regularization is used in PPO across all our benchmarks. The method in Chao et al. (2024) targets earlier normalizing flow models, which differ from the continuous normalizing flows (CNFs) we use. These earlier flows are designed as invertible functions, allowing likelihood computation via the change-of-variable formula and entropy estimation via Monte Carlo methods. However, these techniques do not directly extend to CNFs as used in our work.
>
> **Reference**:
>
> [1] Ding, Shutong, Ke Hu, Zhenhao Zhang, Kan Ren, Weinan Zhang, Jingyi Yu, Jingya Wang, and Ye Shi. "Diffusion-based reinforcement learning via q-weighted variational policy optimization." Advances in Neural Information Processing Systems 37 (2024): 53945-53968.

---

> > ### Author Response · Authors · 2025-11-22
> >
> > > Experimental weakness 1: The experimental evaluation is also fragmented—PolicyFlow is compared with FPO on MuJoCo and PPO on IsaacLab, preventing a unified cross-method assessment.
> >
> > We thank the reviewer for this suggestion. In the revised manuscript, **we have conducted a unified comparison on the MuJoCo Playground benchmark, including FPO, DPPO, PPO, and PolicyFlow,** to provide a more rigorous and comprehensive evaluation and avoid a fragmented experimental assessment.
> >
> > We **retain the original PPO vs. PolicyFlow comparison on IsaacLab** because IsaacLab is actively supported by NVIDIA, provides a well-established and standardized testing pipeline, and is widely used in the robotics community. **FPO was not included in this environment because its open-source code is implemented in JAX, whereas IsaacLab is based on PyTorch, making code migration nontrivial; we have noted this in the revised paper**.
> >
> > ---
> >
> > > Experimental weakness 2: Including all three under identical settings would clarify relative efficiency and expressivity. Furthermore, results are averaged over five seeds without significance testing
> >
> > We thank the reviewer for this comment. In each IsaacLab environment, we ensured that PolicyFlow and PPO use **nearly identical hyperparameter settings**, except for the entropy regularization coefficients $w_b$ and $w_g$. These coefficients were set according to the principle $w_b^{\text{PF}} + w_g^{\text{PF}} = w_g^{\text{PPO}}$. The rationale is that while entropy regularization promotes exploration, it can compete with average return maximization during convergence; following this principle ensures a fair comparison. These tests indicate that hyperparameters tuned for PPO can be effectively transferred to PolicyFlow, which is beneficial for hyperparameter portability.
> >
> > Similarly, in MuJoCo Playground, we kept PPO and PolicyFlow hyperparameters nearly identical, applying the same principle for $w_b$ and $w_g$. FPO, however, appears to be more sensitive to hyperparameters and cannot use the same settings as PPO. Potential reasons include the asymmetric error introduced by its importance ratio approximation and differences in gradient scale due to the flow-matching loss, but verifying these factors is beyond the scope of our current work.
> >
> > Following the reviewer’s suggestion, **we have added variance reporting and computed $p-$values using Welch’s $t-$test** to strengthen the rigor of our results.
> >
> > ---
> >
> >
> > > Experimental weakness 3: The MuJoCo Playground evaluation covers only a moderate subset of tasks (e.g., BallInCup, CheetahRun, FingerTurnHard, ReacherHard) but omits the most challenging or high-dimensional environments such as Humanoid or Walker. This limits claims of scalability to complex continuous-control domains.
> >
> > We thank the reviewer for pointing out the limitation in our benchmark selection. In the revised manuscript, **we have added experiments on the Walker environment and removed the PointMass test**.  We hope this provides a more comprehensive comparison of our method with others.
> >
> > We would also like to note that the IsaacLab benchmarks we use predominantly consist of challenging and high-dimensional environments, which further demonstrate the scalability of our approach.
> >
> > ---
> >
> > > Minor Issues:  some notations is not well defined: Sec. 3. 𝑠,𝑎 is not defined before using. 𝑝_𝜋(𝑠) is never defined. 𝜋∗ is mentioned but not clearly defined.
> >
> > We thank the reviewer for pointing out these notational issues. In response, we have made the following revisions in the manuscript:
> >
> > (1) Added definitions of $s$ and $a$ at the beginning of Section 3.
> >
> > (2) Clarified $p_\pi(s)$ in Line 150.
> >
> > (3) Clarified the meaning of $\pi^*$ in Line 160.
> >
> > These changes should make the notation clearer and more consistent throughout the paper.
> >
> > ---
> >
> > > Q1: Can you clarify whether the interpolation-based importance ratio introduces any noticeable bias? It would help to know if you’ve compared it against an exact or less-approximated version.
> >
> > We thank the reviewer for this question. Indeed, our interpolation-based approximation introduces some bias. However, this bias can be controlled by the PPO clipping parameter $\epsilon$. To further illustrate its effect, **we have added a set of comparative experiments showing how different values of $\epsilon$ influence performance (Sec. 5.3)**.
> >
> > For reference, FPO establishes a connection between the flow-matching loss and the flow model log-likelihood via an ELBO, providing an alternative approximation. But FPO paper does not provide any methods for entropy regularization. And we have included FPO as a baseline in the MuJoCo Playground benchmark.

---

### Official Review · Reviewer_UPw3 · 2025-11-01

**Soundness:** 2
**Presentation:** 3
**Contribution:** 2
**Rating:** 2
**Confidence:** 4

**Summary:**

The paper introduces PolicyFlow, an on-policy reinforcement learning algorithm that replaces the standard Gaussian policy used in PPO with a continuous normalizing flow parameterized by ordinary differential equations. This modification aims to enhance expressiveness for complex or multimodal action distributions. The approach also incorporates a Brownian motion-inspired implicit entropy regularizer to promote exploration without relying on explicit policy entropy terms. The paper compares PolicyFlow against PPO and other flow-based policy optimization methods, such as FPO, reporting similar or slightly improved empirical results on selected benchmarks.

**Strengths:**

- Addresses the expressiveness limitation of Gaussian policies by exploring normalizing flows for policy representation.
- Introduces a Brownian motion-based entropy regularizer to encourage implicit exploration.
- Presents a clear and structured implementation based on PPO.
- Includes runtime and parameter analyses, offering transparency on computational cost.
- Demonstrates engagement with related work, including comparisons to other flow-based methods.

**Weaknesses:**

- The reported empirical results are very close to PPO, providing limited evidence of improvement.
- The additional model complexity and slower runtime are not justified by corresponding performance gains.
- The theoretical connection between the flow-based representation and policy gradient optimization is underdeveloped.
- The motivation for emphasizing FPO comparisons is not well justified relative to the paper’s main objective.
- Benchmark evaluations and variance reporting are incomplete, limiting the strength of the experimental conclusions.

**Questions:**

**Detailed Review:**

The paper proposes PolicyFlow, which replaces the Gaussian policy distribution in PPO with a continuous normalizing flow modeled through ordinary differential equations. The design aims to capture multimodal action distributions and improve policy expressiveness. The use of a Brownian motion-inspired implicit entropy regularizer seeks to enhance exploration without introducing explicit entropy terms in the loss.

Although this approach is conceptually sound, the paper does not provide strong empirical or theoretical evidence that PolicyFlow meaningfully improves over PPO. The observed results in Figures and Tables suggest comparable performance rather than clear gains. The discussion of results should better address why these similarities occur and whether the proposed approach offers other advantages, such as stability or robustness.

The focus on comparison with FPO is noted, but it is unclear why this method receives particular attention when the central comparison should remain with PPO. It would be helpful to clarify the methodological differences that justify this choice. The analysis could also include visualization or theoretical examples that demonstrate how the normalizing flow structure affects action representation or optimization dynamics.

While the Brownian entropy regularizer is an appealing idea, its effect on exploration and stability is not quantified. A comparison with standard entropy regularization would help assess whether it provides genuine benefit or simply replicates existing behavior. Similarly, the discussion on computational efficiency should more clearly address the trade-offs between the added parameterization and any observed gains in performance.

The runtime and parameter analyses are appreciated, as they help contextualize the computational implications of the proposed method. However, since the improvements are small and the training time increases, the practical benefit of adopting PolicyFlow remains uncertain. The lack of variance reporting in Tables 1 and 2 also limits confidence in the consistency of the results.

In summary, the idea of integrating normalizing flows into policy gradient methods is interesting and potentially valuable, but the paper does not yet establish sufficient justification, theoretical insight, or empirical advantage to support its adoption.

**Questions:**

1. What is the specific benefit of using normalizing flows over Gaussian policies in terms of policy expressiveness and action representation?
2. How does the Brownian entropy regularizer contribute to exploration compared with standard entropy regularization?
3. What is the motivation for emphasizing comparisons with FPO, and how do these comparisons support the main claims of the paper?
4. How does PolicyFlow perform in terms of computational cost and sample efficiency relative to PPO?
5. Could the authors include a simple theoretical or illustrative example showing how the flow-based policy captures multimodal actions more effectively?
6. Are the experimental results averaged over multiple runs, and if so, could variance be reported to strengthen the reliability of the findings?
7. What is the runtime difference between **sampling action** from the normalizing flow policy and from the Gaussian policy?
8. Under what conditions does PolicyFlow provide a clear improvement over PPO, and how does this relate to its theoretical formulation within policy gradient optimization?

**Details Of Ethics Concerns:**

None.

---

> ### Author Response · Authors · 2025-11-22
>
> We would like to sincerely thank reviewer UPw3 for the careful and thoughtful review of our manuscript. We greatly appreciate the time and effort you took to thoroughly read our paper and provide detailed feedback. Your insights regarding the empirical results, theoretical clarity, baseline comparisons, and experimental reporting have been extremely valuable. We have carefully considered all your comments and, where appropriate, incorporated additional analyses, ablation studies, and clarifications in the revised manuscript. Your feedback has helped us improve both the clarity and rigor of our work.
>
> ------
>
> > Weakness 1: The reported empirical results are very close to PPO, providing limited evidence of improvement.
>
> We thank the reviewer for the insightful comment regarding the empirical performance relative to PPO. Our intention is to clarify that our contribution is not aimed at outperforming PPO on tasks where Gaussian policies are already nearly optimal, but rather to extend the PPO framework to expressive policy classes—specifically continuous normalizing flows (CNFs)—which cannot be trained using standard PPO due to the intractability of their likelihoods.
>
> In many continuous-control benchmarks, the optimal policy is essentially a deterministic mapping from states to actions. A Gaussian policy with sufficiently small variance already provides an excellent approximation in such settings. Since CNFs strictly generalize Gaussian policies and can represent simple unimodal distributions when appropriate, PolicyFlow naturally converges to policies that behave similarly to Gaussian PPO on these tasks. Thus, the closeness in performance is expected and, in fact, confirms that our method correctly recovers PPO-level performance when Gaussian policies are sufficient.
>
> At the same time, one major advantage of using CNFs is their ability to represent much richer and multimodal action distributions, which enables significantly more diverse exploration behaviors. This benefit becomes evident in environments where multimodality is essential. As demonstrated in the **newly added PointMaze experiments (Fig. 1)**, which were specifically included to showcase the effect of the Brownian regularizer on exploration, PolicyFlow produces substantially more diverse and balanced exploration patterns. Similarly, in the MultiGoal experiments (Fig. 2), **we added comparisons with DPPO as well as existing methods that use heuristic noise injection for entropy regularization**, further highlighting the effectiveness of our approach. In contrast, PPO and existing flow-based baselines tend to collapse to a limited set of modes.
>
> These results highlight that PolicyFlow preserves PPO’s strengths on standard continuous-control tasks while unlocking richer exploration and more expressive policy behaviors when the environment demands them.

---

> ### Author Response · Authors · 2025-11-22
>
> > Weakness2: The additional model complexity and slower runtime are not justified by corresponding performance gains.
>
> We appreciate the reviewer’s concern and agree that, on many standard continuous-control benchmarks, the performance gains from using CNF-based policies may not appear large enough to fully justify the increased model complexity and runtime. This is a valid observation. Indeed, most existing benchmarks do not naturally expose the advantages of expressive generative policies, because their underlying optimal policies are close to deterministic and can already be well approximated by a simple Gaussian distribution.
>
> The main motivation for PolicyFlow is therefore **not to improve performance on these particular benchmarks, but to enable on-policy reinforcement learning for expressive policy classes**, specifically continuous normalizing flows, which are widely used in robotics and visuomotor policy learning but cannot be trained directly with PPO due to intractable likelihoods. **Although generative models such as CNFs and diffusion policies have proven highly effective and robust in robot manipulation and visuomotor tasks**  [1,2,3,4], **they are typically trained only on offline datasets using denoising score matching or flow matching. These training pipelines have reached clear performance bottlenecks, and leveraging reinforcement learning to further improve such policies is widely viewed as a promising next step**.
>
> In this context, extending PPO to support CNF-based policies is an important and timely direction. PPO is already a dominant algorithm in training large-scale policies for LLMs and robot locomotion, and enabling it to optimize expressive generative policies is a capability that the community actively needs, evidenced by recent methods such as FPO and DPPO, which also attempt to address this challenge.
>
> While many standard benchmarks do not reveal the representational benefits of CNFs, tasks that inherently require multimodal behavior do. As demonstrated in the MultiGoal (Fig. 2) and PointMaze (Fig. 1) results, PolicyFlow enables substantially richer and more diverse behaviors than PPO and existing generative-policy baselines, highlighting the practical value of CNFs where multimodality matters.
>
> In summary, although the performance differences on standard benchmarks may appear modest, **the contribution of PolicyFlow lies in providing a principled and efficient framework for training expressive CNF-based policies with PPO, a capability that is increasingly important in modern robotic and visuomotor learning pipelines**.
>
> **Reference**:
>
> [1] Lei, Kun, Huanyu Li, Dongjie Yu, Zhenyu Wei, Lingxiao Guo, Zhennan Jiang, Ziyu Wang, Shiyu Liang, and Huazhe Xu. "RL-100: Performant Robotic Manipulation with Real-World Reinforcement Learning." arXiv preprint arXiv:2510.14830 (2025).
>
> [2] Intelligence, Physical, Kevin Black, Noah Brown, James Darpinian, Karan Dhabalia, Danny Driess, Adnan Esmail et al. "$\pi_ {0.5} $: a Vision-Language-Action Model with Open-World Generalization." arXiv preprint arXiv:2504.16054 (2025).
>
> [3] Gao, Dechen, Boqi Zhao, Andrew Lee, Ian Chuang, Hanchu Zhou, Hang Wang, Zhe Zhao, Junshan Zhang, and Iman Soltani. "Vita: Vision-to-action flow matching policy." arXiv preprint arXiv:2507.13231 (2025).
>
> [4] Chi, Cheng, Zhenjia Xu, Siyuan Feng, Eric Cousineau, Yilun Du, Benjamin Burchfiel, Russ Tedrake, and Shuran Song. "Diffusion policy: Visuomotor policy learning via action diffusion." The International Journal of Robotics Research 44, no. 10-11 (2025): 1684-1704.

---

> > ### Author Response · Authors · 2025-11-22
> >
> > > Weakness 3: The theoretical connection between the flow-based representation and policy gradient optimization is underdeveloped.
> >
> > We appreciate the reviewer’s comment and agree that the manuscript may not have sufficiently clarified the relationship between our flow-based policy representation and policy gradient optimization. To the best of our understanding, there is no inherent theoretical connection between flow-based models and policy gradient methods themselves. **The flow-based representation simply specifies how the policy is parameterized**, while policy gradient optimization can in principle be applied to any differentiable policy class.
> >
> > Most prior work uses policies with tractable likelihoods (e.g., Gaussians) because standard policy gradient methods require evaluating the log-probability of sampled actions. Flow-based models, however, typically do not provide closed-form densities for conditional distributions in the way required by PPO. This is precisely the challenge that PolicyFlow addresses.
> >
> > If the reviewer’s concern is about how the mathematical expressions for log-probabilities change when moving from a Gaussian policy to a CNF-based policy, we agree this point deserves clearer explanation. Equations (5)–(7) in the revised manuscript explicitly describe this transition and illustrate how the log-probability of an action can be computed through the instantaneous change-of-variables formula during the flow integration. **We have further revised the text surrounding these equations to make the motivation and the derivation clearer**, and to help readers understand how PPO can be extended from Gaussian policies to CNF-based policies in a principled manner.
> >
> > We thank the reviewer again for highlighting this point, which helped us improve the clarity of the theoretical presentation.
> >
> > ---
> >
> > > Weakness 4: The motivation for emphasizing FPO comparisons is not well justified relative to the paper’s main objective.
> >
> > > Q3: What is the motivation for emphasizing comparisons with FPO, and how do these comparisons support the main claims of the paper?
> >
> > We thank the reviewer for raising this important point. The core focus of our paper is on training continuous normalizing flows (also known as flow-matching models) with PPO, which is also the main idea behind FPO. A key challenge in this setting is that CNFs do not have an explicit probability density function, making the evaluation of the importance ratio difficult. Both FPO and our method propose ways to approximate this importance ratio, which motivates the comparison with FPO.
> > To address the reviewer’s concern, **we have added a discussion at the beginning of the Experiments section explaining our choice of baselines. In addition to FPO, we also include comparisons with DPPO**.
> >
> > Our method differs from FPO in several ways: (i) **we introduce an entropy regularization technique, (ii) we retain the adaptive learning rate strategy based on KL divergence from PPO, which FPO notes is not feasible in their setup**, and (iii) our approach uses a different approximation for the importance ratio. We have added these clarifications in the Experiments section to better justify our choice of FPO as a baseline.
> >
> > In tests on the MuJoCo Playground benchmark using FPO’s parameter settings, our method achieves results comparable to FPO. However, consistent with recent work FPO++ (https://openreview.net/forum?id=BA6n0nmagi), **we observe that FPO tends to suffer from policy collapse as training progresses**. Potential reasons include the lack of an adaptive learning rate and asymmetric errors introduced by their importance ratio approximation, which may destabilize training when the distribution of positive and negative samples becomes uneven during convergence.

---

> ### Author Response · Authors · 2025-11-22
>
> > Q1: What is the specific benefit of using normalizing flows over Gaussian policies in terms of policy expressiveness and action representation?
>
> Thank reviewer for this important question. **The main benefit of using continuous normalizing flows (CNFs) over Gaussian policies is their ability to represent much richer, potentially multimodal action distributions**. While Gaussian policies are unimodal and thus limited to capturing a single dominant mode, CNFs can model arbitrarily complex, structured, and multimodal behaviors, enabling more expressive policies and richer exploration.
>
> In recent robotics research, it has become increasingly common to represent policies using generative models such as CNFs (also called flow-matching models) or diffusion models. Most existing methods train these generative policies purely through imitation learning (e.g., denoising score matching or flow matching). While effective to some extent, imitation learning faces clear limitations in data collection and performance saturation. As a result, many recent works—including $\pi^* 0.6$, RL100, DPPO, and FPO—are actively exploring how to train generative policies directly with reinforcement learning. Our work aligns with this direction, providing a framework that enables PPO to optimize CNF-based policies.
>
> Regarding the benefits of CNFs over Gaussian policies, the primary advantage lies in expressiveness. **As illustrated in our MultiGoal experiments, CNF-based policies naturally produce richer and more diverse action behaviors that Gaussian policies cannot express**. These representational benefits are particularly important in **contact-rich** or **multimodal** robotic manipulation tasks, where the optimal policy often involves switching among multiple distinct action strategies. This is precisely why generative models have become the preferred choice in visuomotor policy learning and robot manipulation pipelines: their ability to capture complex, multimodal action distributions leads to improved robustness and performance.
>
> In summary, CNFs offer substantially greater policy expressiveness than Gaussian policies, enabling richer exploration and more diverse action behaviors, and our work provides a practical RL framework for training such expressive policies.
>
> ---
>
> > Q2: How does the Brownian entropy regularizer contribute to exploration compared with standard entropy regularization?
>
> We thank the reviewer for this insightful question. As noted, **there is currently no computationally feasible standard entropy regularization for continuous normalizing flows**. While the entropy of a standard Gaussian policy can be computed analytically, flow models defined via ODEs do not have an explicit probability density function, making analytical computation intractable, and Monte Carlo estimation is also challenging.
>
> In order to better address the reviewer’s concern, we have **added new experiments in the MultiGoal (Fig. 2) and PointMaze environments (Fig. 1)**. These experiments compare our Brownian entropy regularizer with existing heuristic methods [1] and demonstrate its effectiveness in encouraging exploration and mitigating mode collapse. We hope these additional results help clarify the benefits of our approach.
>
> **Reference**:
>
> [1] Ding, Shutong, Ke Hu, Zhenhao Zhang, Kan Ren, Weinan Zhang, Jingyi Yu, Jingya Wang, and Ye Shi. "Diffusion-based reinforcement learning via q-weighted variational policy optimization." Advances in Neural Information Processing Systems 37 (2024): 53945-53968.
>
> ---
>
> > Q4: How does PolicyFlow perform in terms of computational cost and sample efficiency relative to PPO?
>
> We thank the reviewer for this question. In fact, a direct comparison between standard PPO and PolicyFlow is not meaningful, as standard PPO cannot train a CNF policy. **Our work can be viewed as an extension of PPO to enable the training of CNF policies**.
>
> If we were to compare the two in terms of computational cost, training a CNF policy generally requires more computation than a Gaussian policy. However, this comes with a significant advantage: CNFs can represent much more complex, potentially multi-modal action distributions, which standard Gaussian policies cannot capture.

---

> ### Author Response · Authors · 2025-11-22
>
> > Q5: Could the authors include a simple theoretical or illustrative example showing how the flow-based policy captures multimodal actions more effectively?
>
> We thank the reviewer for this suggestion. Our MultiGoal toy example (Fig. 2) was specifically designed to illustrate that a flow-based policy can capture multimodal actions. Using our method, the trained policy allows the agent to reach six different goal points from roughly the same starting position. Additionally, **we have added experiments on the PointMaze environment (Fig. 1) to further demonstrate that flow-based policies enable more diverse and comprehensive exploration**. Moreover, **A figure has been added in Page 4 to provide a high-level overview of the algorithm**, illustrating how the updates of the flow's velocity field are influenced by factors such as the RL advantage and Brownian regularization. This figure helps clarify the overall dynamics of the flow-based policy training process.
>
> From a theoretical perspective, one can consider the following illustrative example. Suppose for a given observation there are two actions with equal advantage values. A Gaussian policy would likely output their mean, resulting in an “average” action. In contrast, a flow-based policy associates each action with a distinct latent variable $z$. One can interpret this as augmenting the original observation with $z$, so the two actions correspond to different augmented observations. This mechanism allows the flow-based policy to avoid collapsing multiple optimal actions into a single average action.
>
> ---
> > Weakness 5: Benchmark evaluations and variance reporting are incomplete, limiting the strength of the experimental conclusions.
>
> > Q6: Are the experimental results averaged over multiple runs, and if so, could variance be reported to strengthen the reliability of the findings?
>
> We thank the reviewer for pointing out this issue. All experiments in IsaacLab and MuJoPlay Ground were run with 5 different random seeds, and the results are averaged over these runs. We have updated the manuscript to report the variance of our experimental results. In addition, we have performed statistical analysis using Welch’s t-test to compute $p$-values, further strengthening the reliability and robustness of our findings.
>
> ---
>
> > Q7: What is the runtime difference between sampling action from the normalizing flow policy and from the Gaussian policy?
>
> We thank the reviewer for this question. Sampling from a CNF policy involves simulating an ODE, where the network models the velocity field $\dot{x} = f(x)$. At inference time, a sample $x_0$ is drawn from a Gaussian, and the ODE is simulated to produce $x_1$, which is the output of the flow policy. This simulation requires multiple evaluations of the network to compute the velocity field at each integration step.
>
> In contrast, a Gaussian policy typically models the mean and variance directly, which can be seen as a fixed mapping $y = f(x)$. At inference, the network only needs to be evaluated once to sample an action.
>
> Therefore, under the same network architecture and number of parameters (as in our experiments with MLPs), sampling from a flow-based policy is approximately $N$times slower than from a Gaussian policy. Here, $N$ depends on the specific ODE solver used during inference (determining the number $P$ of network evaluations per step) and the number of discrete time steps $T$, giving $N = P \times T$.
>
> ---
>
> > Q8: Under what conditions does PolicyFlow provide a clear improvement over PPO, and how does this relate to its theoretical formulation within policy gradient optimization?
>
> We thank the reviewer for this question. **PolicyFlow can be seen as an extension of PPO** that enables training policies represented by CNFs, which is not possible with standard PPO since it cannot handle policies without an explicit probability density function. **In fact, a CNF policy is capable of representing Gaussian policies as a special case, implying that CNFs can implicitly model a larger set of probability distributions**.
>
> We understand that the reviewer is particularly interested in the conditions under which replacing a Gaussian policy with a CNF policy leads to performance improvements. Prior work and empirical evidence in robotic manipulation tasks suggest that generative policies such as CNFs are better suited for **contact-rich tasks with non-unique paths, where capturing multimodal action distributions is critical for effective performance**.
>
> Additionally, standard policy gradient optimization using Monte Carlo estimation relies on evaluating the probability of each sampled action. While CNF policies can model complex distributions, they do not explicitly model the PDF, making this evaluation intractable. To address this challenge, we propose an approximation method for the PPO importance ratio, enabling effective optimization with CNF policies.

---

### Official Review · Reviewer_PmDD · 2025-11-01

**Soundness:** 3
**Presentation:** 3
**Contribution:** 3
**Rating:** 8
**Confidence:** 4

**Summary:**

This paper introduces PolicyFlow, a novel on-policy reinforcement learning algorithm designed to leverage the expressive power of Continuous Normalizing Flows (CNFs) for policy representation within a PPO-like framework.  They show how to avoids the need for full ODE simulation and backpropagation during the policy update and how to implement the entropy regularization in a smart way in the generative policy situation by imitating the Brownian motion. They finally show improved performance compared to the traditional PPO and related flow-based PPO method.

**Strengths:**

1. The authors demonstrate how to bypass the computationally expensive full ODE simulation and backpropagation typically required when using Neural ODE-based policies with a PPO objective. Their key insight is to use an efficient approximation of the importance ratio, enabling stable on-policy training without the standard computational bottlenecks.
2. the paper introduces a lightweight "Brownian regularizer" to enhance behavioral diversity and mitigate mode collapse.

**Weaknesses:**

1. How is the initial flow matching model for the method in this paper obtained? What is the impact of the initial model's performance on the overall method?

2. The target distribution of the flow-based policy changes dynamically during training, yet the objective function samples only a single t from the path at each step. Could this, due to the varying sample weights (different values of A) for each t along the path, prevent the model from learning an effective distribution?

3. A sensitivity analysis experiment should be conducted on the parameters used in the algorithm, e.g, Brownian regularizer weight wb.

**Questions:**

1, since the first-order approximation of the importance ratio may introduce bias into the gradient estimate, have you empirically assessed its impact? For example, did you observe any degradation in performance or training instability as the PPO clipping range ϵ increases, which would amplify the approximation error?

2, could you clarify the sampling strategy for tk (Algorithm 1, lines 15–16)? In the final experiments, did you use uniform sampling U[0, 1], and if so, did it perform better than sampling from the discrete simulation time points?

---

> ### Author Response · Authors · 2025-11-22
> **Add ablation studies on model initialization method, clippping range $\epsilon$ and time sampling strategies**
>
> We sincerely thank the reviewer for the thoughtful and encouraging assessment of our work. We greatly appreciate the reviewer’s clear summary, recognition of the key strengths, particularly our efficient approximation of the importance ratio and the Brownian regularizer, and the constructive questions and suggestions that help us further improve the paper.
>
> > Weakness 1: How is the initial flow matching model for the method in this paper obtained? What is the impact of the initial model's performance on the overall method?
>
> We thank the reviewer for this insightful question. The initial flow matching model is not trained at all. It is simply obtained by randomly initializing all network parameters. For all experiments, we use the commonly adopted **Glorot (Xavier) initialization** [1], which samples the weights from a Gaussian distribution with an appropriate variance; the output layer is initialized to zeros following standard practice.
>
> To address the reviewer’s concern, we **added an additional ablation in the revision paper (Sec. 5.4) to analyze how different initialization strategies affect our method**. The results show that while initialization can influence the early-stage optimization behavior, it does not change the overall performance trend nor the main conclusions of our approach.
>
> Reference:
>
> [1] Glorot, X. & Bengio, Y. (2010). Understanding the difficulty of training deep feedforward neural networks.
>
> ---
>
> > Weakness 2: The target distribution of the flow-based policy changes dynamically during training, yet the objective function samples only a single t from the path at each step. Could this, due to the varying sample weights (different values of A) for each t along the path, prevent the model from learning an effective distribution?
>
> > Q2: Ccould you clarify the sampling strategy for tk (Algorithm 1, lines 15–16)? In the final experiments, did you use uniform sampling U[0, 1], and if so, did it perform better than sampling from the discrete simulation time points?
>
> We thank the reviewer for this detailed question. As formulated in our final objective (Eq. (12) and Eq. (15)), the expectation over time $t$ is estimated using Monte Carlo sampling. For each sample in a mini-batch, $t$ is independently drawn from a uniform distribution over [0,1]. While it is true that the effective sample weight can vary due to different advantage values $A$ at each $t$, the Monte Carlo estimator is unbiased, and its variance decreases as the number of samples increases, allowing the model to learn an effective distribution.
>
>
> To further address the reviewer’s concern regarding the use of a single $t$ per sample, we conducted **additional experiments comparing (Sec. 5.4)**: (i) uniform sampling from [0,1], (ii) sampling from discrete simulation time points, and (iii) sampling multiple $t$ values per state-action sample. The results, included in the revised paper, show that even with a single $t$ state-action sample, the model can learn effectively, and **the choice of sampling strategy has only a minor impact on the final performance**.
>
> ---
>
> > Weakness 3: A sensitivity analysis experiment should be conducted on the parameters used in the algorithm, e.g, Brownian regularizer weight $w_b$.
>
> We thank the reviewer for raising this important point. Compared with PPO, the primary differences in our method are the introduction of time-$t$ sampling and the Brownian regularizer weight $w_b$. All other hyperparameters remain identical to those in PPO, and tuning practices from PPO generally transfer directly to our approach. To address the reviewer’s concern, we have **added an additional experiment that analyzes how the parameter $w_b$ influences exploration (Fig. 1)**. Furthermore, because the clipping range affects the approximation error, we have also **added an ablation study on this factor (Sec. 5.3)**.
>
> ---
>
> > Q1: Since the first-order approximation of the importance ratio may introduce bias into the gradient estimate, have you empirically assessed its impact? For example, did you observe any degradation in performance or training instability as the PPO clipping range ϵ increases, which would amplify the approximation error?
>
> We thank the reviewer for this question. We conducted **additional experiments to analyze the effect of the PPO clipping range $\epsilon$ on performance (Sec. 5.3)**, addressing the reviewer’s concern regarding the first-order approximation of the importance ratio. Our results indicate a clear trade-off: **a smaller $\epsilon$ reduces the approximation error and keeps the policy KL divergence between updates small, which stabilizes training but slows down policy improvement, reducing sample efficiency. Conversely, a larger $\epsilon$ allows larger updates but increases the approximation error, which can also degrade sample efficiency**. These findings are included in the revised paper, providing practical guidance for choosing $\epsilon$.

---

### Author Response · Authors · 2025-11-22

We would like to sincerely thank the reviewers for their careful and constructive feedback. We greatly appreciate the professional perspective on the theoretical rigor, experimental completeness, and clarity of result presentation, as well as the suggestions regarding estimation bias, algorithmic sample efficiency, and exploration, all of which have helped us improve this work. We have carefully addressed all reviewer concerns point by point, added the necessary experiments requested by the reviewers, and made appropriate revisions to the paper to make the work more rigorous, reliable, and broadly useful to the community.

We would also like to emphasize the significance of our contribution. **This paper extends PPO from training traditional Gaussian models that approximate real-world Probability Density Functions (PDFs) to training generative models such as Continuous Normalizing Flows (CNFs, also called Flow-Matching model). CNFs are strictly more expressive than Gaussian distributions and can represent complex multimodal distributions**. As expected, in fully observable continuous control tasks, we obtain performance comparable to PPO, since the optimal policy in such tasks is often close to a deterministic mapping, which a low-noise Gaussian can well approximate while still ensuring a certain level of exploration.

**The practical significance of enabling PPO to train generative models like CNFs lies in embodied intelligence research**. Prior work has shown the superior performance of generative models in robotic manipulation which is contact-rich task, largely due to their ability to capture complex, multimodal distributions. However, these models are mostly trained via imitation learning, such as denoising score matching or flow matching, which is increasingly facing performance bottlenecks. There is growing interest in incorporating reinforcement learning to overcome these limitations and push the field to a new level. **Our work provides a concrete option to train CNF-based (flow-matching) generative models using RL**.

**Additionally, to facilitate more effective exploration, we design a novel entropy regularizer (Brownian Regularizer) that encourages agents to explore while avoiding mode collapse**. Prior to our work, defining a computationally feasible entropy regularizer for CNF-type models has been challenging due to the difficulty of efficiently computing their entropy. Inspired by Brownian motion, we propose a simple yet elegant method: regularizing the velocity field so that it flows toward low-probability regions, which correspond exactly to the score function. This approach is both conceptually interesting and practically effective, as demonstrated by our experimental results.

Finally, we sincerely thank the reviewers again for providing us with the opportunity to further improve and clarify our work. All the changes we made are **highlighted in blue in the revised paper**, and we hope our work meaningfully advance the field. We would also like to note that while the reviewers mentioned the absence of code, we now have a pre-release version available (https://github.com/PolicyFlow2026/PolicyFlow) and plan to release the full code alongside the paper.

---

> ### Author Response · Authors · 2025-12-03
>
> Below we summarize the key concerns raised by the reviewers and the concrete actions we have taken in the rebuttal and revised manuscript.
>
> ---
>
> ### **1. Main Reviewer Concerns**
>
> 1. **Interpolation-based importance-ratio approximation**
>
>    Reviewers asked whether the approximation introduces bias, how large the bias may be, and whether PPO clipping (ε) can effectively control it.
>
> 2. **Brownian Regularizer (implicit entropy)**
>
>    Reviewers requested clearer motivation and quantitative comparisons against standard entropy bonuses or noise-based exploration.
>
> 3. **Experimental coverage and baselines**
>
>    Concerns included baseline completeness, diversity of tasks, fairness of comparisons, variance reporting, and statistical significance.
>
> 4. **Sensitivity to hyperparameters and sampling strategies**
>
>    Multiple reviewers asked for ablations on initialization, time-sampling distributions, Brownian weight, and clipping range (ε) .
>
> 5. **Theoretical clarity and reproducibility**
>
>    Requests included clearer notation, more explicit derivations, discussion of approximation error, and availability of code / hyperparameter tables.
>
> ---
>
> ### **2. How We Addressed These Concerns**
>
> 1. **Approximation error analysis**
>
>    Added a new analysis in Appendix A and a dedicated ablation in Sec. 5.3 on PPO clipping (ε), showing how bias–stability–efficiency trade-offs behave in practice and providing tuning guidelines.
>
> 2. **Quantitative evaluation of Brownian Regularizer**
>
>    Added experiments on MultiGoal comparing Brownian regularization with standard entropy and noise injection, demonstrating improved multimodal exploration and reduced mode collapse. Added experiments on PointMaze showing the affect of Brownian regularizer on exploration.
>
> 3. **Expanded and unified experimental evaluation**
>
>    * Added a unified comparison in *MuJoCo Playground* (PPO, FPO, DPPO, PolicyFlow).
>    * Added a new Walker task in MuJoCo Playground to broaden task diversity.
>    * Reported mean ± std over 5 seeds and Welch t-test p-values for all major tables.
>
> 4. **Comprehensive ablations and sensitivity analyses**
>
>    * Added initialization ablations (e.g., Xavier) in Sec. 5.4.
>    * Added analyses on different time-sampling schemes (uniform, discrete, multi-t) and clipping-range sensitivity.
>    * Provided recommended defaults and practical tuning guidance.
>
> 5. **Improved theoretical clarity and reproducibility**
>
>    * Clarified derivations around Eq. (5)–(7) and cleaned up notation (s, a, pπ(s), π*, etc.).
>    * Added discussion of interpolation-related approximation bounds in the appendix.
>    * Provided a code pre-release link and detailed hyperparameter tables.
>
> ---
>
> ### **3. Clarifying the Contribution Scope**
>
> Our goal is **not** to claim universal superiority over Gaussian-PPO on deterministic continuous-control benchmarks. Rather, the contribution is to **extend the strengths of PPO to a more expressive class of policies, namely continuous normalizing flows (CNFs), also known as Flow-Matching Model**, enabling stable on-policy training of multimodal, high-capacity generative policies.
>
> We show that PolicyFlow is particularly beneficial on tasks requiring **multimodal behavior and richer exploration**, and we strengthened both theory and experiments to clarify this applicability.
>
> In particular, the **Brownian Regularizer** offers an **elegant, lightweight, and highly effective mechanism** for entropy regularization in flow-matching models, providing strong incentives for exploration and reducing mode collapse.

---

### Meta-Review · Area_Chair_s1Ea · 2026-01-04

**Summary:**

The reviewers' concerns centred around the following:
- whether the experimental evaluation is strong enough
- whether the regulariser is sufficiently justified experimentally (experiments are needed since he paper offers no theory on regularisation or generalisation)
- whether the importance sampling correction works in practice (it is an approximation)

**Reviewer Concerns:**

1. (addressed) The authors provided additional experiments.
2. (addressed) They illustrate the performance of the Brownian noise, although I see some conflation between its effect on generalisation (i.e. as a regulariser) and its effect on exploration
3. (addressed) They provided an analysis of the error introduced by the importance sampling correction

**Reviewer Scores:**

reviewer PmDD (score 8) -> still score 8
reviewer UPw3 (score 2) -> score 4 (because I think the main concerns about what the paper was trying to do were sufficiently addressed)
reviewer gsX4 (Score 4) -> still 4 (because, while the additional evaluations are welcome, they are not super-exhaustive)
reviewer P1yA (score 4) -> still 4 (same reason)

---

### Decision · Program_Chairs · 2026-01-26

Accept (Poster)